# A Conditional Independence Test in the Presence of Discretization

## Abstract

Testing conditional independence (CI) has many important applications, such as Bayesian network learning and causal discovery. Although several approaches have been developed for learning CI structures for observed variables, those existing methods generally fail to work when the variables of interest can not be directly observed and only discretized values of those variables are available. For example, if $X_1$, $\tilde{X}_2$ and $X_3$ are the observed variables, where $\tilde{X}_2$ is a discretization of the latent variable $X_2$, applying the existing methods to the observations of $X_1$, $\tilde{X}_2$ and $X_3$ would lead to a false conclusion about the underlying CI of variables $X_1$, $X_2$ and $X_3$. Motivated by this, we propose a CI test specifically designed to accommodate the presence of discretization. To achieve this, a bridge equation and nodewise regression are used to recover the precision coefficients reflecting the conditional dependence of the latent continuous variables under the nonparanormal model. An appropriate test statistic has been proposed, and its asymptotic distribution under the null hypothesis of CI has been derived. Theoretical analysis, along with empirical validation on various datasets, rigorously demonstrates the effectiveness of our testing methods.

## 1 Introduction

Independence and conditional independence (CI) are fundamental concepts in statistics. They are leveraged for exploring queries in statistical inference, such as sufficiency, parameter identification, adequacy, and ancillarity [9]. They also play a central role in emerging areas such as causal discovery [18], graphical model learning, and feature selection [36]. Tests for CI have attracted increasing attention from both theoretical and application sides.

Formally, the problem is to test the CI of two variables $X_{j_1}$ and $X_{j_2}$ given a random vector (a set of other variables) $\boldsymbol{Z}$. In statistical notation, the null hypothesis is written as $H_0 : X_{j_1} \perp X_{j_2} \mid \boldsymbol{Z}$, where $\perp$ denotes "independent from." The alternative hypothesis is written as $H_1 : X_{j_1} \not\perp X_{j_2} \mid \boldsymbol{Z}$, where $\not\perp$ denotes "dependent with." The null hypothesis implies that once $\boldsymbol{Z}$ is known, the values of $X_{j_1}$ provide no additional information about $X_{j_2}$, and vice versa. Different tests have been designed to handle different scenarios, including Gaussian variables with linear dependence [37, 25, 22, 26] and non-linear dependence [16, 38, 31, 27, 1] (*For detailed related work, please refer to App. D*).

Given observations of $X_{j_1}$, $X_{j_2}$, and $\boldsymbol{Z}$, the CI can be effectively tested with existing methods. However, in many scenarios, accurately measuring continuous variables of interest is challenging due to limitations in data collection. Sometimes the data obtained are approximations represented as discretized values. For example, in finance, variables such as asset values cannot be measured and are binned into ranges for assessing investment risks (e.g., sell, hold, and strong buy) [7, 8]. Similarly, in mental health, anxiety levels are often assessed using scales like the GAD-7, which categorizes

responses into levels such as mild, moderate, or severe [23, 17]. In the entertainment industry, the quality of movies is typically summarized through viewer ratings [29, 10].

When discretization is present, existing CI tests can fail to determine the CI of underlying continuous variables. This issue arises because existing CI tests treat discretized observations as observations of continuous variables, leading to incorrect conclusions about their CI relationships. More precisely, the problem lies in the discretization process, which introduces new discrete variables. Consequently, *although the intent is to test the CI of the underlying continuous variables, what is actually being tested is the CI involving a mix of both continuous and newly introduced discrete variables*. In general, this CI relationship is inconsistent with the one among the underlying continuous variables.

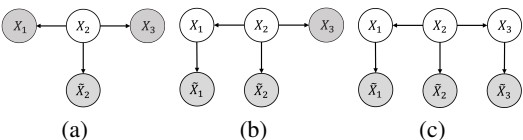

(a)     (b)     (c)

Figure 1: We illustrate different data generative processes with causal graphical models. The discretization process introduces new discrete variables which are denoted with a tilde ($\sim$).

As illustrated in Fig. 1, we show different data-generative processes using causal graphical models [24] in the presence of discretization. A gray node indicates an observable variable, while a white node indicates a latent variable. Variables denoted by $X_j$ (without a tilde $\sim$) represent continuous variables, which may not be observed; while variables denoted by $\tilde{X}_j$ represent observed discretized variables derived from $X_j$ due to discretization. In Fig. 1(a), $X_2$ is latent, and only its discrete counterpart $\tilde{X}_2$ is observed. In this case, rather than observing $X_1$, $X_2$, and $X_3$, we only observe $X_1$, $\tilde{X}_2$, and $X_3$. Existing CI methods use these observations to test **whether** $X_1 \perp X_3 \mid \{X_2\}$, but what is actually being tested is **whether** $X_1 \perp X_3 \mid \{\tilde{X}_2\}$. In fact, according to the *causal Markov condition* [30], , it can be inferred from Fig. 1(a) that $X_1 \perp X_3 \mid \{X_2\}$ and $X_1 \not\perp X_3 \mid \{\tilde{X}_2\}$. This mismatch leads to existing CI methods, that employ observations to check the CI relationships between $X_1$ and $X_3$ given $X_2$, to reach incorrect conclusions. Due to the same reason, checking the CI also fails in Fig 1(b) and Fig 1(c).

In this paper, we design a CI test specifically for handling the presence of discretization. An appropriate test statistic for the CI of latent continuous variables, based solely on discretized observations, is derived. The key is to build connections between the discretized observations and the parameters needed for testing the CI of the latent continuous variables. To achieve this, we first develop bridge equations that allow us to estimate the covariance of the underlying continuous variables with discretized observations. Then, we leverage a *node-wise regression* [5] to derive appropriate test statistics for CI relationships from the estimated covariance. By assuming that the continuous variables follow a Gaussian distribution, we can derive the asymptotic distributions of the test statistics under the null hypothesis of CI. The major contributions of our paper include that

- We develop a CI test for ensuring accurate analysis in scenarios where data has been discretized, which are common due to limitations in data collection or measurement techniques, such as in financial analysis and healthcare.

- Our CI test can handle various scenarios including 1). Both variables $X_{j_1}$ and $X_{j_2}$ are discretized 2). Both variables $X_{j_1}$ and $X_{j_2}$ are continuous. 3). One of the variables $X_{j_1}$ or $X_{j_2}$ is discretized.

- We compare our test with the existing methods on both synthetic and real-world datasets, confirming that our method can effectively estimate the CI of the underlying continuous variables and outperform the existing tests applied on the discretized observations.

## 2   DCT: A CI Test in the Presence of Discretization

**Problem Setting**   Consider a set of independent and identically distributed (i.i.d.) $p$-dimensional random vectors, denoted as $\tilde{\boldsymbol{X}} = (X_1, X_2, \ldots, \tilde{X}_j, \ldots, \tilde{X}_p)^T$. In this set, some variables, indicated by a tilde ($\sim$), such as $\tilde{X}_j$, follow a discrete distribution. For each such variable, there exists a corresponding latent Gaussian random variable $X_j$. The transformation from $X_j$ to $\tilde{X}_j$ is governed by an unknown monotone nonlinear function $g_j$. This function, $g_j : \mathcal{X} \to \tilde{\mathcal{X}}$, maps the continuous domain of $X_j$ onto the discrete domain of $\tilde{X}_j$, such that $\tilde{X}_j = g_j(X_j)$ for each observation. Given $n$ observations $\{\tilde{\boldsymbol{x}}^1, \tilde{\boldsymbol{x}}^2, \ldots, \tilde{\boldsymbol{x}}^n\}$ randomly sampled from $\tilde{\boldsymbol{X}}$, specifically, for each variable $X_j$, there

exists a constant vector $\mathbf{d} = (d_1, \ldots, d_M)$ characterized by strictly increasing elements such that

$$\tilde{x}_j^i = \begin{cases} 1 & 0 < g_j(x_j^i) < d_1 \\ m & d_{m-1} < g_j(x_j^i) < d_m \\ M & g_j(x_j^i) > d_m \end{cases} \tag{1}$$

This model is also known as the nonparanormal model [20]. The cardinality of the domain after discretization is at least 2 and smaller than infinity. Our goal is to assess both conditional and unconditional independence among the variables of the vector $\boldsymbol{X} = (X_1, X_2, \ldots, X_j, \ldots, X_p)^T$. In our model, we assume $\boldsymbol{X} \sim N(0, \Sigma)$, $\Sigma$ only contain 1 among its diagonal, i.e., $\sigma_{jj} = 1$ for all $j \in [1, \ldots, p]$. One should note this assumption is *without loss of generality*. We provide a detailed discussion of our assumption in App. A.8.

**Preliminary Framework of DCT**    To develop an independence test, one needs to design a test statistic that can reflect the dependence relation and be calculated from observations. Next, it is essential to derive the underlying distribution of this statistic under the null hypothesis that the tested variables are conditionally (or unconditionally) independent. By calculating the value of the test statistic from observations and determining if this statistic is likely to be sampled from the derived distribution (i.e., calculating the *p-value* and comparing it with the significance level $\alpha$), we can decide if the null hypothesis should be rejected.

Our objective is to deduce the independence and CI relationships within the original multivariate Gaussian model, based on its discretized observations. In the context of a multivariate Gaussian model, this challenge is directly equivalent to constructing statistical inferences for its covariance matrix $\boldsymbol{\Sigma} = (\sigma_{j_1, j_2})$ and its precision matrix $\boldsymbol{\Omega} = (\omega_{j,k}) = \boldsymbol{\Sigma}^{-1}$ [3]. The covariance matrix $\boldsymbol{\Sigma}$ captures the pairwise covariances between variables, while the precision matrix $\boldsymbol{\Omega}$ (also known as the concentration matrix) provides information about the CI between variables. Specifically, the entry $\omega_{j,k}$ in the precision matrix is related to the partial correlation coefficient between variables $X_j$ and $X_k$, which can be used to test whether these variables are conditionally independent given some other variables. Technically, we are interested in two things: (1) the calculation of the covariance $\hat{\sigma}_{j_1, j_2}$ and the precision coefficient (or the partial correlation coefficient) $\hat{\omega}_{j,k}$, serving as the estimation of $\sigma_{j_1, j_2}$ and $\omega_{j,k}$ respectively (in this paper, a variable with a hat indicates its estimation); and (2) the derivation of the distribution of $\hat{\sigma}_{j_1, j_2} - \sigma_{j_1, j_2}$ and $\hat{\omega}_{j,k} - \omega_{j,k}$ under the null hypothesis of independence and CI.

In the subsequent section, 1). we first introduce ***bridge equations*** to address the estimation challenge of the covariance $\sigma_{j_1, j_2}$; 2). we proceed to derive the distribution of $\hat{\sigma}_{j_1, j_2} - \sigma_{j_1, j_2}$, demonstrating it is ***asymptotically normal***; 3). utilizing ***nodewise regression***, we establish the relationship between the covariance matrix $\boldsymbol{\Sigma}$ and the precision matrix $\boldsymbol{\Omega}$, where the regression parameter $\beta_{j,k}$ acts as an effective surrogate for $\omega_{j,k}$. Leveraging the distribution of $\hat{\sigma}_{j_1, j_2} - \sigma_{j_1, j_2}$, we further illustrate that $\hat{\beta}_{j,k} - \beta_{j,k}$ is also ***asymptotically normal***.

## 2.1    Design *Bridge Equation* for Test Statistics

**Estimating Covariance with Bridge Equations**    The bridge equation establishes a connection between the underlying covariance $\sigma_{j_1, j_2}$ of two continuous variables $X_{j_1}$ and $X_{j_2}$ with the observations. When in the presence of discretization, the discrete transformations make the sample covariance matrix based on $\tilde{\boldsymbol{X}}$ inconsistent with the covariance matrix of $\boldsymbol{X}$. To obtain the estimation $\hat{\sigma}_{j_1, j_2}$ of $\sigma_{j_1, j_2}$, the bridge equation is leveraged. In general, its form is as follows.

$$\hat{\tau}_{j_1, j_2} = T(\sigma_{j_1, j_2}; \hat{\boldsymbol{\Lambda}}), \tag{2}$$

where $\sigma_{j_1, j_2}$ is the covariance needed to be estimated, $\hat{\tau}_{j_1, j_2}$ is a statistic that can also be estimated from observations, and $\hat{\boldsymbol{\Lambda}}$ is a set of additional parameters required by the function $T(\cdot)$. The specific form of the function $T(\cdot)$ will be derived later. Both $\hat{\tau}_{j_1, j_2}$ and $\hat{\boldsymbol{\Lambda}}$ should be able to be calculated purely relying on observations. *Then, given the calculated $\hat{\tau}_{j_1, j_2}$ and $\hat{\boldsymbol{\Lambda}}$, $\hat{\sigma}_{j_1, j_2}$ can be obtained by solving the bridge equation $\hat{\tau}_{j_1, j_2} = T(\sigma_{j_1, j_2}; \hat{\boldsymbol{\Lambda}})$.* As a result, the covariance matrix $\boldsymbol{\Sigma}$ of $\boldsymbol{X}$ can be estimated, which contains information about both unconditional independence and CI (which can be derived from its inverse).

To estimate the covariance of a latent multivariate Gaussian distribution, we need to design appropriate $\hat{\tau}_{j_1, j_2}$, $\hat{\boldsymbol{\Lambda}}$, and $T(\cdot)$. Notably, bridge equations have to be designed to handle all three possible cases:

136 C1. both observed variables are discretized; C2. one variable is continuous while the other is
137 discretized; and C3. both variables remain continuous. We will show that cases C1 and C2 can be
138 merged into a single form of bridge equation with different parameters and a binarization operation
139 applied to the observations. Our bridge equations are presented in Def. 2.2, Def. 2.3, and Def. 2.4.

140 **Bridge Equations for *Discretized and Mixed Pairs***    Let us first address the challenging cases
141 where both observed variables are discretized or where one variable is continuous while the other
142 is discretized. In general, different bridge equations would need to be designed to handle each case
143 individually. *However, in our analysis, we provide a unified bridge equation that is applicable to both*
144 *cases.* This is achieved by binarizing the observed variables, thereby unifying both cases into a binary
145 case. As some information may be lost in the binarization process, this unification may require more
146 examples compared to using tailored bridge functions for each specific case. Developing specific
147 bridge equations for each case to improve sample efficiency is left in future work.

148 Intuitively, for the original continuous variable $X_j$, binarization separates it into two parts based on
149 a boundary $h_j$: the part for $X_j$ larger than $h_j$ and the part for $X_j$ smaller than $h_j$. In this case, we can
150 estimate the boundary by calculating the proportion of $X_j$ that exceeds the boundary. In the scenario
151 of two variables where the threshold $h_{j_1}$ and $h_{j_2}$ divide the space into four regions, the proportions of
152 these areas are influenced by the covariance $\sigma_{j_1,j_2}$, which connects the relation between the binarized
153 variables with the latent covariance. This approach allows us to initially estimate the threshold $h_{j_1}$,
154 $h_{j_2}$ of a pair of variables, followed by estimating the covariance $\sigma_{j_1,j_2}$.

155 Let $\mathbb{P}_n Z$ denote the average of a random variable $Z$ given $n$ i.i.d. observation of $Z$ and $E[Z]$ as the
156 true mean of $Z$, $\mathbb{P}$ as the probability and $\hat{P}$ as the empirical probability. We then define the boundary
157 $h_j$ as follows: for any single discretized variable $\tilde{X}_j$, there exists a constant $c_j$ such that:

$$\mathbb{1}\{\tilde{x}_j^i > E[\tilde{X}_j]\} = \mathbb{1}\{g_j(x_j^i) > c_j\} = \mathbb{1}\{x_j^i > h_j\},$$

158 where $h_j = g_j^{-1}(c_j)$. Specifically, $h_j$ is the boundary in the original continuous domain to determine
159 if the discretized observation $\tilde{X}_k$ is larger than its mean. When the continuous variable $X_j$ follows
160 a normal distribution, there is a relation $\mathbb{P}(\tilde{X}_j > E[\tilde{X}_j]) = 1 - \Phi(h_j)$, where $\Phi$ is the cumulative
161 distribution function (cdf) of a standard normal distribution. We then provide the following definition:

162 **Definition 2.1.** The estimated boundary can be expressed as $\hat{h}_j = \Phi^{-1}(1 - \hat{\tau}_j)$, where $\hat{\tau}_j = $
163 $\sum_{i=1}^n \mathbb{1}_{\{\tilde{x}_j^i > \mathbb{P}_n \tilde{X}_j\}}/n$, serving as the estimation of $\mathbb{P}(\tilde{X}_j > E[\tilde{X}_j])$.

164 Let $\bar{\Phi}(z_1, z_2; \rho) = \mathbb{P}(Z_1 > z_1, Z_2 > z_2)$, where $(Z_1, Z_2)^T$ follows a bivariate normal distribution
165 with mean zero, variance one and covariance $\rho$. We define

$$\tau_{j_1,j_2} = \mathbb{P}(\tilde{x}_{j_1}^i > E[\tilde{X}_{j_1}], \tilde{x}_{j_2}^i > E[\tilde{X}_{j_2}]) = \bar{\Phi}(h_{j_1}, h_{j_2}; \sigma_{j_1,j_2}). \tag{3}$$

166 That is, the proportion of discretized variables larger than their mean can be expressed as a function
167 of underlying covariance. This equation serves as the key of estimating latent covariance based on the
168 discretized observations. Specifically, we can substitute those true parameters with their estimation
169 and construct the bridge equation to get the estimated covariance:

170 **Definition 2.2** (Bridge Equation for A Discretized-Variable Pair). For discretized variables $\tilde{X}_{j_1}$ and
171 $\tilde{X}_{j_2}$, the bridge equation is defined as:

$$\hat{\tau}_{j_1,j_2} = \hat{P}(\tilde{X}_{j_1} > \mathbb{P}_n \tilde{X}_{j_1}, \tilde{X}_{j_2} > \mathbb{P}_n \tilde{X}_{j_2}) = \frac{1}{n}\sum_{i=1}^n \mathbb{1}_{\{\tilde{x}_{j_1}^i > \mathbb{P}_n \tilde{X}_{j_1}, \tilde{x}_{j_2}^i > \mathbb{P}_n \tilde{X}_{j_2}\}} = T(\sigma_{j_1,j_2}; \{\hat{h}_{j_1}, \hat{h}_{j_2}\}),$$

and the function $T(\sigma_{j_1,j_2}; \{\hat{h}_{j_1}, \hat{h}_{j_2}\}) := \bar{\Phi}(\hat{h}_{j_1}, \hat{h}_{j_2}; \sigma) = \int_{x_1 > \hat{h}_{j_1}} \int_{x_2 > \hat{h}_{j_2}} \phi(x_{j_1}, x_{j_2}; \sigma) dx_{j_1} dx_{j_2},$

172 where $\phi$ is the probability density function of a bivariate normal distribution, $\hat{h}_{j_1}, \hat{h}_{j_2}$ can be simply
173 calculated using Def. 2.1.

174 Following the same intuition, we can directly apply the same bridge equation to estimate the co-
175 variance of mixed pairs. The only difference is there is no need to estimate the boundary $\hat{h}_j$ for the
176 continuous variable. Instead, we can incorporate its true mean of zero into the equation.

**Definition 2.3** (Bridge Equation for A Continuous-Discretized-Variable Pair)**.** For one continuous variable $X_{j_1}$ and one discretized variable $\tilde{X}_{j_2}$, the bridge function is defined as follows:

$$\hat{\tau}_{j_1,j_2} = \hat{P}(X_{j_1} > 0, \tilde{X}_{j_2} > \mathbb{P}_n \tilde{X}_{j_2}) = \frac{1}{n} \sum_{i=1}^{n} \mathbb{1}_{\{x_{j_1}^i > 0, \tilde{x}_{j_2}^i > \mathbb{P}_n \tilde{X}_{j_2}\}} = T(\sigma_{j_1,j_2}; \{0, \hat{h}_{j_2}\}),$$

and the function $T(\cdot)$ has the same form of Def. 2.2.

**A Bridge Equation for A Continuous-Variable Pair**   When there is no discretized transformation, the sample covariance of $X_{j_1}$ and $X_{j_2}$ provides a consistent estimation. In this context, the function $T$ acts merely as an identity mapping.

**Definition 2.4** (A Bridge Equation for A Continuous-Variable Pair)**.** For two continuous variables $X_{j_1}$ and $X_{j_2}$ , the bridge equation is defined as:

$$\hat{\tau}_{j_1,j_2} := \hat{\sigma}_{j_1,j_2} = \frac{1}{n} \sum_{i=1}^{n} x_{j_1}^i x_{j_2}^i - \frac{1}{n} \sum_{i=1}^{n} x_{j_1}^i \frac{1}{n} \sum_{i=1}^{n} x_{j_2}^i = T(\sigma_{j_1,j_2}; \emptyset).$$

For two continuous variables $X_{j_1}$ and $X_{j_2}$, the analytic solution of the estimated covariance can be simply obtained using Def. 2.4.

**Calculation of Estimated Covariance**   For the continuous case, the analytic solution of $\hat{\sigma}_{j_1,j_2}$ can be simply obtained using Def. 2.4. For the cases involving the discretized variable as proposed in Def. 2.2 and Def. 2.3, we can rely on the property that variance $\Sigma$ only contains 1 among the diagonal, which implies the covariance $\sigma_{j_1,j_2}$ should vary from $-1$ to $1$. Thus, we can calculate the estimated covariance by solving the objective

$$\min_{\sigma_{j_1,j_2}} ||\hat{\tau}_{j_1,j_2} - T(\sigma_{j_1,j_2}; \{\hat{h}_{j_1}, \hat{h}_{j_2}\})||^2 \quad s.t. -1 < \sigma_{j_1,j_2} < 1. \tag{4}$$

The $\hat{\tau}_{j_1,j_2}$ is a one-to-one mapping with calculated $\hat{\sigma}_{j_1,j_2}$, $\hat{h}_{j_1}$ and $\hat{h}_{j_2}$, which is proved in App. A.2

## 2.2   Unconditional Independence Test

The estimation of covariance $\hat{\sigma}_{j_1,j_2}$ can be effectively solved using the designed bridge equation. Now, we focus on deriving the distribution of $\hat{\sigma}_{j_1,j_2} - \sigma_{j_1,j_2}$. These results is used as an unconditional independence test in the presence of the discretization. Moreover, Thm. 2.5, Lem. 2.6, Lem. 2.7 and Lem. 2.8 will be leveraged in the derivation process of the CI test in Section 2.3. The detailed derivation steps for both unconditional test and CI test are relatively intricate, therefore, we will provide a general intuition. For a complete derivation, please refer to the App. A.3.

Assume we are interested in the true parameter $\theta_0$. We denote $\hat{\theta}$ as its estimation which is close to $\theta_0$, and $f(\theta)$ is a continuous function. By leveraging Taylor expansion, we have

$$f(\hat{\theta}) = f(\theta_0) + f'(\theta_0)(\hat{\theta} - \theta_0), \tag{5}$$

which directly constructs the relationship between the estimated parameter with the true one. Re-arrange the term, we get $\hat{\theta} - \theta_0 = (f(\hat{\theta}) - f(\theta_0))/f'(\theta_0)$. If the denominator is a constant and the numerator can be expressed as a sum of i.i.d samples, we can see $\hat{\theta} - \theta_0$ will be asymptotically normal according to the central limit theorem [35].

Let $\psi_{\hat{\theta}} = [f_{\hat{\theta}}^1(\cdot), f_{\hat{\theta}}^2(\cdot), f_{\hat{\theta}}^3(\cdot)]^T$ contains a group of functions parameterized by $\hat{\theta}$ (For discretized pairs, $\hat{\theta} = (\hat{\sigma}_{j_1,j_2}, \hat{h}_{j_1}, \hat{h}_{j_2})$). Define $\mathbb{P}_n \psi_{\hat{\theta}}$ as sample mean of these functions evaluated at $n$ sample points. Similarly, $\mathbb{P}_n \psi_{\hat{\theta}} \psi_{\hat{\theta}}^T$ is defined as sample mean of the outer product $\psi_{\hat{\theta}} \psi_{\hat{\theta}}^T$. The notation $P\psi_{\hat{\theta}} := E\mathbb{P}_n \psi_{\hat{\theta}}$ denotes the expectations of the functions in $\psi_{\hat{\theta}}$. Furthermore, let $\psi_{\hat{\theta}}'$ denote the derivative of the functions contained in $\psi_{\hat{\theta}}$. We now provide the main result of derived distribution $\hat{\sigma}_{j_1,j_2} - \sigma_{j_1,j_2}$ under the hull hypothesis that test pairs are independent.

**Theorem 2.5** (Independence Test)**.** *In our settings, under the null hypothesis that two observed variables indexed with $j_1$ and $j_2$ are statistically independent under our framework, i.e., $\sigma_{j_1,j_2} = 0$, the independence can be tested using the statistic*

$$\hat{\sigma}_{j_1,j_2} = T^{-1}(\hat{\tau}_{j_1,j_2}; \hat{\theta}).$$

*This statistic is approximated to follow a normal distribution, as detailed below:*

$$\hat{\sigma}_{j_1,j_2} \overset{approx}{\sim} N\left(0, \frac{1}{n}((\mathbb{P}_n\psi'_{\hat{\theta}})^{-1}\mathbb{P}_n\psi_{\hat{\theta}}\psi_{\hat{\theta}}^T(\mathbb{P}_n\psi'^T_{\hat{\theta}})^{-1})_{1,1}\right), \tag{6}$$

*where the specific form of $\psi_{\hat{\theta}}$ are presented in Lem. 2.6,Lem. 2.7 and Lem. 2.8.*

We now provide the specific forms of $\psi_{\hat{\theta}}$. Since the variables being tested for independence can be both discretized, only one being discretized, or neither being discretized. This results in different forms of $\psi_{\hat{\theta}}$ consequently differs across these scenarios. Let $Z_{j_1}$ and $Z_{j_2}$ be any two random variables indexed by $j_1$ and $j_2$. Let $\hat{\sigma}^i_{j_1,j_2} = z^i_{j_1} \cdot z^i_{j_2} - \mathbb{P}_n Z_{j_1} \cdot \mathbb{P}_n Z_{j_2}$ denote the sample covariance based on a $i$-th pairwise observation of the variables $Z_{j_1}$ and $Z_{j_2}$. Let $\hat{\tau}^i_{j_1} = \mathbb{1}_{\{z^i_{j_1} > \mathbb{P}_n Z_{j_1}\}}$ and $\hat{\tau}^i_{j_2} = \mathbb{1}_{\{Z^i_{j_2} > \mathbb{P}_n Z_{j_2}\}}$, each calculated based on $i$-th observations of the variables $Z_{j_1}$ and $Z_{j_2}$, respectively. Let $\hat{\tau}^i_{j_1,j_2}$ be $\hat{\tau}^i_{j_1} \cdot \hat{\tau}^i_{j_2}$. We further denote $\bar{\Phi}(\cdot) = 1 - \Phi(\cdot)$. The different forms of $\psi_{\hat{\theta}}$ that arise in different cases are defined as follows:

**Lemma 2.6.** *($\psi_{\hat{\theta}}$ for A Continuous-Variable Pair). For two continuous variables $X_{j_1}$ and $X_{j_2}$,*

$$\psi_{\hat{\theta}} := \hat{\sigma}^i_{j_1,j_2} - \hat{\sigma}_{j_1,j_2}. \tag{7}$$

**Lemma 2.7** ($\psi_{\hat{\theta}}$ for A Discretized-Variable Pair). *For discretized variables $\tilde{X}_{j_1}$ and $\tilde{X}_{j_2}$,*

$$\psi_{\hat{\theta}} := \begin{pmatrix} \hat{\tau}^i_{j_1,j_2} - T(\hat{\sigma}_{j_1,j_2}; \{\hat{h}_{j_1}, \hat{h}_{j_2}\}) \\ \hat{\tau}^i_{j_1} - \bar{\Phi}(\hat{h}_{j_1}) \\ \hat{\tau}^i_{j_2} - \bar{\Phi}(\hat{h}_{j_2}) \end{pmatrix}. \tag{8}$$

**Lemma 2.8** ($\psi_{\hat{\theta}}$ for A Continuous-Discretized-Variable Pair). *For one discretized variable $\tilde{X}_{j_2}$ and one continuous variable $X_{j_1}$,*

$$\psi_{\hat{\theta}} := \begin{pmatrix} \hat{\tau}^i_{j_1,j_2} - T(\hat{\sigma}_{j_1,j_2}; \{0, \hat{h}_{j_2}\}) \\ \hat{\tau}^i_{j_1} - \bar{\Phi}(\hat{h}_{j_2}) \end{pmatrix}. \tag{9}$$

Derivation of forms of $\psi_{\hat{\theta}}$ for different cases and their corresponding distribution defined in Eq (6) can be found in App. A.4, App. A.5, App. A.6. Up to this point, our discussion has been confined to the case of covariance $\sigma_{j_1,j_2}$, the indicator of unconditional independence. In the next section, we will present the results of our CI test.

## 2.3 Conditional Independence (CI) Test

To construct a CI test of our model, we are interested at two things: calculation of the estimated precision coefficient $\hat{\omega}_{j,k}$ and the derivation of the corresponding distribution $\hat{\omega}_{j,k} - \omega_{j,k}$. In the following, we first build $\beta_{j,k}$, which is obtained using nodewise regression and show it serves as a surrogate of testing for $\omega_{j,k} = 0$, we then construct the formulation of $\hat{\beta}_{j,k} - \beta_{j,k}$ as the combination of formulation of $\hat{\sigma}_{j_1,j_2} - \sigma_{j_1,j_2}$ and show it will also be asymptotically normal.

**Nodewise Regression for CI** To utilize covariance for testing CI, it is necessary to establish a relationship between the estimated covariance and a metric capable of reflecting CI. To achieve this, we employ the nodewise regression which effectively builds the connection between covariance and precision matrix. Suppose we can access observations $\{\boldsymbol{x}^1, \boldsymbol{x}^2, \dots, \boldsymbol{x}^n\}$ from latent continuous variables $\boldsymbol{X} = (X_1, \dots, X_p) \sim N(0, \boldsymbol{\Sigma})$, nodewise regression will do regression on every dimension with all other dimensions as predictors.

$$x^i_{j_1} = \sum_{j_1 \neq j_2} x^i_{j_2}\beta_j + \epsilon^i_{j_1}. \tag{10}$$

It can be shown that there are deterministic relationships between the regression coefficients and the covariance and precision matrices of $\boldsymbol{X}$, as illustrated below and proved in App. A.7.1.

$$\beta_j = \boldsymbol{\Sigma}^{-1}_{-j-j}\boldsymbol{\Sigma}_{-jj} \in \mathbb{R}^{p-1}, \quad \beta_{j,k} = -\frac{\omega_{j,k}}{\omega_{j,j}}, \quad j \neq k, \tag{11}$$

where $\boldsymbol{\Sigma}_{-j-j}$ is the submatrix of $\boldsymbol{\Sigma}$ without $j$th column and $j$th row, and the $\boldsymbol{\Sigma}_{-jj}$ is the vector of $j$th column without $j$th row. $\beta_{j,k} \in \mathbb{R}$ is the surrogate of $\omega_{j,k}$ to capture the independence relationship of $X_j$ with $X_k$ conditioning on other variables. We can use Def. 2.2, Def. 2.3 and Def. 2.4 to get the estimation $\hat{\boldsymbol{\Sigma}}_{-j-j}$ and $\hat{\boldsymbol{\Sigma}}_{-jj}$ and thus get the estimation $\hat{\beta}_j$.

**Statistical Inference for** $\beta_{j,k}$    Nodewise regression offers a robust solution for the estimation problem. A pertinent inquiry pertains to the construction of the distribution of $\hat{\beta}_j - \beta_j$. It is crucial to recognize that the distribution of $\hat{\sigma}_{j_1,j_2} - \sigma_{j_1,j_2}$ is already established. Therefore, if we can conceptualize $\hat{\beta}_j - \beta_j$ as a linear combination of $\hat{\sigma}_{j_1,j_2} - \sigma_{j_1,j_2}$, the problem is directly solved, i.e., the $\hat{\beta}_j - \beta_j$ is linear combination of dependent Gaussian variables. The underlying relationship between these variables is as follows:

$$\hat{\beta}_j - \beta_j = -\hat{\mathbf{\Sigma}}_{-j-j}^{-1}\left((\hat{\mathbf{\Sigma}}_{-j-j} - \mathbf{\Sigma}_{-j-j})\beta_j - (\hat{\mathbf{\Sigma}}_{-jj} - \mathbf{\Sigma}_{-jj})\right).$$

The derivation is provided in App. A.7.2. For ease of notation, we further express the distribution of the difference between the estimated covariance and the true covariance as

$$\hat{\sigma}_{j_1,j_2} - \sigma_{j_1,j_2} = \frac{1}{n}\sum_{i=1}^{n}\xi_{j_1,j_2}^i. \tag{12}$$

The specific form of $\xi_{j_1,j_2}^i$ is given in App. A.4, A.5, A.6 respectively for different cases. For notational convenience, we express $\hat{\mathbf{\Sigma}}_{-j-j} - \mathbf{\Sigma}_{-j-j} = \frac{1}{n}\sum_{i=1}^{n}\Xi_{-j,-j}^i$ and $\hat{\mathbf{\Sigma}}_{-jj} - \mathbf{\Sigma}_{-jj} = \frac{1}{n}\sum_{i=1}^{n}\Xi_{-j,j}^i$, where $\xi_{j_1,j_2}$ is the element of the matrix $\Xi$ at the position indexed by $(j_1, j_2)$. We now propose the statistic and its asymptotic distribution for the CI test in the following theorem.

**Theorem 2.9** (Conditional Independence test)**.** *In our settings, under the null hypothesis that $X_j$ and $X_k$ are conditional statistically independent given a set of variables $\mathbf{Z}$, i.e., $\beta_{j,k} = 0$, the statistic*

$$\hat{\beta}_{j,k} = (\hat{\mathbf{\Sigma}}_{-j-j}^{-1}\hat{\mathbf{\Sigma}}_{-jj})_{[k]}, \tag{13}$$

*where $[k]$ denotes the element corresponding to the variable $X_k$ in $\hat{\mathbf{\Sigma}}_{-j-j}^{-1}\hat{\mathbf{\Sigma}}_{-jj}$. The statistic $\hat{\beta}_{j,k}$ has the asymptotic distribution:*

$$\hat{\beta}_{j,k} \sim N(0, a^{[k]^T}\frac{1}{n^2}\sum_{i=1}^{n}vec(B_{-j}^i)vec(B_{-j}^i)^T a^{[k]}),$$

*where $B^i = \begin{bmatrix} \Xi_{-j,-j}^i \\ \Xi_{-j,j}^i \end{bmatrix}$,    $a_l^{[k]} = \begin{cases} \left(\hat{\mathbf{\Sigma}}_{-j-j}^{-1}\right)_{[k],l}, & for\ l \in \{1,\dots,p-1\} \\ \sum_{q=1}^{n}\left(\hat{\mathbf{\Sigma}}_{-j-j}^{-1}\right)_{[k],l}\left(\tilde{\beta}_j\right)_q, & for\ l \in \{p,\dots,p^2-p\} \end{cases}$*

*and $\tilde{\beta}_j$ is $\beta_j$ whose $\beta_{j,k} = 0$.*

In practice, we can plug in the estimation of regression parameter $\hat{\beta}_j$ and set $\hat{\beta}_{j,k} = 0$ as the substitution of $\tilde{\beta}_j$ to calculate the variance and do the CI test. Specifically, we can obtain the $\hat{\beta}_{j,k}$ using Eq. (13) where the estimated covariance terms can be calculated by solving the bridge equation Eq. 2. Under the null hypothesis that $\beta_{j,k} = 0$ (conditional independence), we can take the calculated $\hat{\beta}_{j,k}$ into the distribution defined in Thm. 2.9 and obtain the p-value. If the p-value is smaller than the predefined significance level $\alpha$ (normally set at 0.05), we will infer the tested pairs are conditionally dependent; otherwise, we do not. The detailed derivation of the Thm. 2.9 can be found in App. A.7.2.

# 3   Experiments

We applied the proposed method DCT to synthetic data to evaluate its practical performance and compare it with Fisher-Z test [14] (for all three data types) and Chi-Square test [15] (for discrete data only) as baselines. Specifically, we investigated its Type I and Type II error and its application in causal discovery. The experiments investigating its robustness, performance in denser graphs and effectiveness in a real-world dataset can be found in App. C.

## 3.1   On the Effect of the Cardinality of Conditioning Set and the Sample Size

Our experiment investigates the variations in Type I and Type II error (1 minus power) probabilities under two conditions. In the first scenario, we focus on the effects of modifying the sample size, denoted as $n = (100, 500, 1000, 2000)$, while conditioning on a single variable. In the second, the sample size is held constant at 2000, and we vary the cardinality of the conditioning set, represented

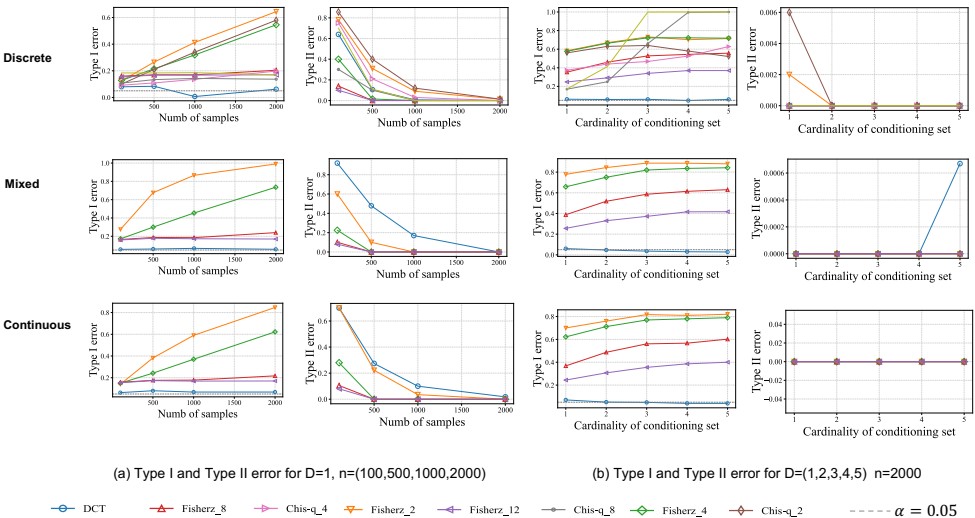

(a) Type I and Type II error for D=1, n=(100,500,1000,2000)   (b) Type I and Type II error for D=(1,2,3,4,5) n=2000

DCT — Fisherz_8 — Chis-q_4 — Fisherz_2 — Fisherz_12 — Chis-q_8 — Fisherz_4 — Chis-q_2 — - - - $\alpha = 0.05$

Figure 2: Comparison of results of Type I and Type II error ($1 -$ power) for all three types of tested data (continuous, mixed, discrete) and different number of samples and cardinality of conditioning set. The suffix attached to a test's name denotes the cardinality of discretization; for example, "Fsherz_4" signifies the application of the Fsher-z test to data discretized into four levels. Chi-square test is only applicable for the discrete case.

as $D = (1, 2, \ldots, 5)$. It is assumed that every variable within this conditioning set is effective, i.e., they influence the CI of the tested pairs. We repeat each test 1500 times.

We use $Y, W$ to denote the variables being tested and use $Z$ to denote the variables being conditioned on. The discretized versions of the variables are denoted with a tilde symbol (e.g., $\tilde{Z}$). For both conditions, we evaluate three distinct types of observations of tested variables: continuous observations for both variables $(Y, W)$, discrete observations for both variables $(\tilde{Y}, \tilde{W})$ and a mixed type $(\tilde{Y}, W)$. The variables in the conditioning set will always be discretized observations $(\tilde{Z})$.

To see how well the derived asymptotic null distribution approximates the true one, we verify if the probability of Type I error aligns with the significance level $\alpha$ preset in advance. We generate true continuous multivariate Gaussian data $Y, W$ from $Z_i$ (single $i = 1$ for the first scenario, and summed over $n$ for the second), structured as $a_i Z_i + E$ and $\sum_{i=1}^{n} a_i Z_i + E$, where $a_i$ is sampled from $U(0.5, 1.5)$ and $E$ follows a standard normal distribution, independent of all other variables. This ensures $Y \perp\!\!\!\perp W | Z$. The data are then discretized into $K = (2, 4, 8, 12)$ levels, with boundaries randomly set based on the variable range. The first column in Fig. 2 (a) (b) shows the resulting probability of Type I errors at the significance level $\alpha = 0.05$ compared with other methods.

A good test should have as small a probability of Type II error as possible, i.e., a larger power. To test the power of our DCT, we generate the continuous multivariate Gaussian data $Z_i$ from $Y, W$; constructed as $Z_i = a_i Y + b_i W + E$, where $a_i, b_i$ are sampled from $U(0.5, 1.5)$ and $E$ follows a standard normal distribution independent with all others, i.e., $Y \not\perp\!\!\!\perp W | Z$. The same discretization approach is applied here. The second column in Fig. 2 (a) (b) shows the Type II error with the changing number of samples and cardinality of the conditioning set compared with other methods.

From Fig. 2 (a), we note that the Type I error rates with our derived null distribution are well-approximated at 0.05 across all three data types in both scenarios. In contrast, other testing methods show significantly higher Type I error rates, increasing with the number of samples and the size of the conditioning set. This indicates that such methods are more prone to erroneously concluding that tested variables are conditionally dependent. Additionally, while alternative tests demonstrate considerable power with smaller sample sizes, our approach requires a sample size of 2000 to achieve satisfactory power, particularly in mixed and continuous cases. A possible explanation for this phenomenon is that our method binarizes discretized data, which may not effectively utilize all observations. This aspect warrants further investigation in future research. Moreover, our test shows remarkable stability in response to changes in the number of conditioning sets.

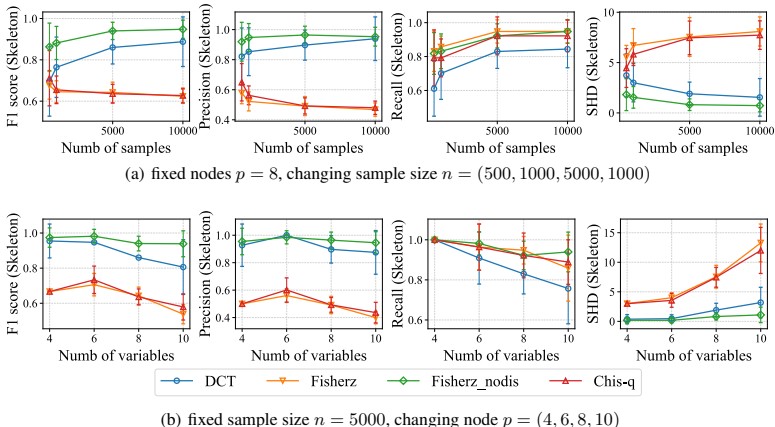

(a) fixed nodes $p = 8$, changing sample size $n = (500, 1000, 5000, 1000)$

(b) fixed sample size $n = 5000$, changing node $p = (4, 6, 8, 10)$

Figure 3: Experiment result of skeleton discovery on synthetic data for changing sample size (a) and changing number of nodes (b). Fisherz_nodis is the Fisher-z test applied to original continuous data. We evaluate $F_1$ (↑), Precision (↑), Recall (↑) and SHD (↓).

## 3.2 Application in Causal Discovery

Causal discovery aims to recover the true causal structure from the data. Constraint-based causal discovery methods like the PC algorithm [30] rely on testing CI from observations to discover causal graphs. However, in the presence of discretization, failures in testing CI leads to false conclusions about causal graphs. To evaluate the efficacy of the DCT, we construct causal graphs utilizing the Bipartite Pairing (BP) model as detailed in [2], with the number of edges being one fewer than the number of nodes. The detailed generation process is provided in App. B due to limited space. Our experiment is divided into two scenarios: (a) fixed data samples $n = 5000$, with changing number of nodes $p = (4, 6, 8, 10)$; (b) fixed number of nodes $p = 8$ and changing sample sizes $n = (500, 1000, 5000, 10000)$.

Comparative analysis is conducted using the PC algorithm integrated with different testing methods. Specifically, we compare DCT against the Fisher-Z test applied to discretized data, the chi-square test, and the Fisher-Z test on original continuous data, the latter serving as a theoretical upper bound for comparison. Since the PC algorithm can only return a completed partially directed acyclic graph (CPDAG), we use the same orientation rules [11] implemented by Causal-DAG [6] to convert a CPDAG into a DAG. We evaluate both the undirected skeleton and the directed graph using criteria such as structural Hamming distance (SHD), F1 score, precision, and recall. For each setting, we run 10 graph instances with different seeds and report the mean and standard deviation of skeleton discovery in Fig. 3, and DAG in Fig. 4 in App B.

According to the result, DCT exhibits performance nearly on par with the theoretical upper bound across metrics such as F1 score, precision, and Structural Hamming Distance (SHD) when the number of variables ($p$) is small and the sample size ($n$) is large. Despite a decline in performance as the number of variables increases with a smaller sample size, DCT significantly outperforms both the Fisher-Z test and the Chi-square test. Notably, in almost all settings, the recall of DCT is lower than that of the baseline tests, which is a reasonable outcome *since these tests tend to infer conditional dependencies, thereby retaining all edges given the discretized observations.* For instance, a fully connected graph, would achieve a recall of 1.

## 4 Conclusion

In this paper, we present a new testing method tailored for scenarios commonly encountered in real-world applications, where variables, though inherently continuous, are only observable in their discretized forms. Our method distinguishes itself from existing CI tests by effectively mitigating the misjudgment introduced by discretization and accurately recovering the CI relationships of latent continuous variables. We substantiate our approach with theoretical results and empirical validation, underscoring the effectiveness of our testing methods.

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

472 $(\sigma_{j_1,j_2}, h_{j_1}, h_{j_2}), (\sigma_{j_1,j_2}, h_{j_2}), (\sigma_{j_1,j_2})$ *respectively.*

473 *Proof* We first focus on the most challenging one where both variables are discrete. According to
474 the law of large numbers, for the estimated boundary $\hat{h}_{j_1}$ and $\hat{h}_{j_2}$ whose calculations are defined as

475 $\hat{h}_j = \Phi^{-1}(1 - \hat{\tau}_j)$, we should have

$$n \to \infty, \quad \hat{\tau}_j = \frac{1}{n} \sum_{i=1}^{n} \mathbb{1}_{\{\tilde{x}_j^i > \mathbb{P}_n \tilde{X}_j\}} \xrightarrow{p} \mathbb{P}(\tilde{X}_j > E[\tilde{X}_j]). \tag{14}$$

476 Recall the definition $\mathbb{P}(\tilde{X}_j > E[\tilde{X}_j]) = 1 - \Phi(h_j)$, according to continuous mapping theorem [34],
477 as long as the function $\Phi^{-1}(1 - \cdot)$ is continuous, we should have $\hat{h}_j \xrightarrow{p} h_j$. And thus $\hat{h}_{j_1} \xrightarrow{p} h_{j_1}$,
478 $\hat{h}_{j_2} \xrightarrow{p} h_{j_2}$.

479 We have $\hat{\tau}_{j_1,j_2} = \bar{\Phi}(\hat{h}_{j_1}, \hat{h}_{j_2}, \hat{\sigma}_{j_1,j_2})$ and the estimation $\hat{\sigma}_{j_1,j_2}$ can be obtained through solving the
480 function. Similarly, we also have

$$n \to \infty, \quad \hat{\tau}_{j_1,j_2} = \frac{1}{n} \sum_{i=1}^{n} \mathbb{1}_{\{\tilde{x}_{j_1}^i > \mathbb{P}_n \tilde{X}_{j_1}\}} \mathbb{1}_{\{\tilde{x}_{j_2}^i > \mathbb{P}_n \tilde{X}_{j_2}\}} \xrightarrow{p} \mathbb{P}(\tilde{x}_{j_1}^i > E[\tilde{X}_{j_1}], \tilde{x}_{j_2}^i > E[\tilde{X}_{j_2}]) = \tau_{j_1,j_2}.$$
$$\tag{15}$$

481 Similarly, according to the continuous mapping theorem, we have $\hat{\sigma}_{j_1,j_2} \xrightarrow{p} \sigma_{j_1,j_2}$. Thus, the
482 parameter $(\hat{\sigma}_{j_1,j_2}, \hat{h}_{j_1}, \hat{h}_{j_2}) \xrightarrow{p} (\sigma_{j_1,j_2}, h_{j_1}, h_{j_2})$.

483 Apparently, the result above could easily extend to the mixed case where we fix $\hat{h}_1 = h_1 = 0$. Using
484 the same procedure, we should have $(\hat{\sigma}_{j_1,j_2}, \hat{h}_{j_2}) \xrightarrow{p} (\sigma_{j_1,j_2}, h_{j_2})$.

485 For the continuous case whose estimated variance is calculated as $\hat{\sigma}_{j_1,j_2} = \frac{1}{n} \sum_{i=1}^{n} x_{j_1}^i x_{j_2}^i -$
486 $\frac{1}{n} \sum_{i=1}^{n} x_{j_1}^i \frac{1}{n} \sum_{i=1}^{n} x_{j_2}^i.$, according to law of large numbers, we should have

$$n \to \infty, \quad \hat{\sigma}_{j_1,j_2} = \frac{1}{n} \sum_{i=1}^{n} x_{j_1}^i x_{j_2}^i - \frac{1}{n} \sum_{i=1}^{n} x_{j_1}^i \frac{1}{n} \sum_{i=1}^{n} x_{j_2}^i \xrightarrow{p} E(X_{j_1} X_{j_2}) - E(X_{j_1}) E(X_{j_2}) = \sigma_{j_1,j_2}.$$
$$\tag{16}$$

### 487 A.2 Proof of one-to-one mapping between $\hat{\tau}_{j_1,j_2}$ with $\hat{\sigma}_{j_1,j_2}$

488 **Lemma A.2.** *For any fixed $\hat{h}_{j_1}$ and $\hat{h}_{j_2}$, $T(\sigma_{j_1,j_2}; \{\hat{h}_{j_1}, \hat{h}_{j_2}\})$ =*
489 $\int_{x_1 > \hat{h}_{j_1}} \int_{x_2 > \hat{h}_{j_2}} \phi(x_{j_1}, x_{j_2}; \sigma) dx_{j_1} dx_{j_2}$, *is a strictly monotonically increasing function on*
490 $\sigma \in (-1, 1)$.

491 *Proof* To prove the lemma, we just need to show the gradient $\frac{\partial T(\sigma_{j_1,j_2}; \{\hat{h}_{j_1}, \hat{h}_{j_2}\}}{\partial \sigma} > 0$ for $\sigma \in (-1, 1)$.

$$\frac{\partial T(\sigma_{j_1,j_2}; \{\hat{h}_{j_1}, \hat{h}_{j_2}\}}{\partial \sigma} == \frac{1}{2\pi \sqrt{(1 - \sigma^2)}} \exp\left(-\frac{(\hat{h}_{j_1}^2 - 2\sigma \hat{h}_{j_1} \hat{h}_{j_2} + \hat{h}_{j_2}^2)}{2(1 - \sigma^2)}\right), \tag{17}$$

492 which is obviously positive for $\sigma \in (-1, 1)$. Thus, we have one-to-one mapping between $\hat{\tau}_{j_1 j_2}$ with
493 the calculated $\hat{\sigma}_{j_1,j_2}$ for fixed $\hat{h}_{j_1}$ and $\hat{h}_{j_2}$.

### 494 A.3 Proof of Thm. 2.5

495 In this section, we provide the proof of Thm. 2.5, which utilizes a regular statistical tool: Z-estimator
496 [33]. Specifically, we are interested in the parameter $\theta$ and we have it estimation $\hat{\theta}$. Let $\boldsymbol{x_1}, \ldots, \boldsymbol{x_n}$
497 are sampled from some true distribution $P$, we can construct the function characterized by the
498 parameter $\theta$ related the $\boldsymbol{x}$ as $\psi_\theta(\boldsymbol{x})$. As long as we have $n$ observations, we can construct the function
499 as follows

$$\Psi_n(\theta) = \frac{1}{n} \sum_{i=1}^{n} \psi_\theta(\boldsymbol{x}_i) = \mathbb{P}_n \psi_\theta. \tag{18}$$

500 We further specify the form

$$\Psi(\theta) = \int \psi_\theta(\boldsymbol{x}) d\boldsymbol{x} = P \psi_\theta. \tag{19}$$

501 Assume the estimator $\hat{\theta}$ is a zero of $\Psi_n$, i.e., $\Psi_n(\hat{\theta}) = 0$ and will converge in probability to $\theta_0$, which
502 is a zero of $\Psi$, i.e., $\Psi(\theta_0) = 0$. Expand $\Psi_n(\hat{\theta})$ in a Taylor series around $\theta_0$, we should have

$$0 = \Psi_n(\hat{\theta}) = \Psi_n(\theta_0) + (\hat{\theta} - \theta_0) \Psi_n'(\theta_0) + \frac{1}{2}(\hat{\theta} - \theta_0) \Psi_n''(\theta_0). \tag{20}$$

Rearrange the equation above, we have

$$
\begin{aligned}
\hat{\theta} - \theta_0 &= -\frac{\Psi_n(\theta_0)}{\Psi'_n(\theta_0) + \frac{1}{2}(\hat{\theta} - \theta_0)\Psi''_n(\theta_0)} \\
&= -\frac{\frac{1}{n}\sum_{i=1}^n \psi_\theta(\boldsymbol{x}_i)}{\Psi'_n(\theta_0) + \frac{1}{2}(\hat{\theta} - \theta_0)\Psi''_n(\theta_0)}.
\end{aligned}
\tag{21}
$$

According to the central limit theorem, the numerator will be asymptotic normal with variance $P\psi^2_{\theta_0}/n$ as the mean $\Psi(\theta_0) = 0$ is zero. The first term of denominator $\Psi'_n(\theta_0)$ will converge in probability to $\Psi'(\theta_0)$ according to the law of large numbers. The second term $\hat{\theta} - \theta_0 = o_P(1)$. [1] As long as the denominator converges in probability and the numerator converges in distribution, according to Slusky's lemma, we have

$$
\sqrt{n}(\hat{\theta} - \theta_0) \rightsquigarrow N\left(0, \frac{P\psi^2_{\theta_0}}{(P\psi'_{\theta_0})^2}\right).
\tag{22}
$$

Extend into the high-dimensional case we should have

$$
\hat{\theta} - \theta_0 = -(\Psi'_n(\theta_0))^{-1}\Psi_n(\theta_0),
\tag{23}
$$

where the second order term is omitted, further assume the matrix $P\psi'_{\theta_0}$ is invertible, we have

$$
\sqrt{n}(\hat{\theta} - \theta_0) \rightsquigarrow N\left(0, (P\psi'_{\theta_0})^{-1}P\psi_{\theta_0}\psi^T_{\theta_0}(P\psi'^T_{\theta_0})^{-1}\right),
\tag{24}
$$

Specifically, in our case $\theta_0 = (\sigma_{j_1,j_2}, \boldsymbol{\Lambda})$, where $\boldsymbol{\Lambda}$ is another parameter set influencing the estimation of $\sigma_{j_1,j_2}$ (will discuss case in case in later proof). In the practical scenario, we only have access to the estimated parameter $\hat{\theta}$ and the empirical distribution $\mathbb{P}_n$, thus we have

$$
\hat{\sigma}_{j_1,j_2} - \sigma_{j_1,j_2} \overset{\text{approx}}{\sim} N\left(0, ((\mathbb{P}_n\psi'_{\hat{\theta}})^{-1}\mathbb{P}_n\psi_{\hat{\theta}}\psi^T_{\hat{\theta}}(\mathbb{P}_n\psi'^T_{\hat{\theta}})^{-1})_{1,1}\right).
\tag{25}
$$

Under the null hypothesis of independent, $\sigma_{j_1,j_2=0}$. We provide the proof that $\hat{\theta} \overset{p}{\to} \theta_0$ of our case in App. A.1. Thus, $\mathbb{P}_n\psi_{\hat{\theta}}$, the function parameterized by $\hat{\theta}$, should also converge in $\mathbb{P}_n\psi_{\hat{\theta}_0}$ when $n \to \infty$. Besides, by the law of large numbers, $\mathbb{P}_n\psi_{\hat{\theta}_0}$ will converge to $P\psi_{\hat{\theta}_0}$. Thus, the equation above will converge to Eq. (24) when $n \to \infty$.

## A.4 Derivation of Lem. 2.7

Let's first focus on the most challenging case where both variables are discretized observations and our interested parameter will include $\hat{\theta} = (\hat{\sigma}_{j_1,j_2}, \hat{h}_{j_1}, \hat{h}_{j_2})$ (Although we only care about the distribution of $\hat{\sigma}_{j_1,j_2} - \sigma_{j_1,j_2}$, the estimation of boundary $\hat{h}_{j_1}$ and $\hat{h}_{j_2}$ will influence the estimation of $\hat{\sigma}_{j_1,j_2}$, thus we need to consider all of them).

The next step will be to *construct an appropriate criterion function $\psi$ such that $\Psi_n(\hat{\theta}) = \mathbf{0}$*. Given $n$ observations $\{\tilde{\boldsymbol{x}}^1, \tilde{\boldsymbol{x}}^2, \ldots, \tilde{\boldsymbol{x}}^n\}$, which are discretized version of $\{\boldsymbol{x}^1, \boldsymbol{x}^2, \ldots, \boldsymbol{x}^n\}$ we should have

$$
\Psi_n(\hat{\theta}) = \begin{pmatrix} \Psi_n(\hat{\sigma}_{j_1,j_2}) \\ \Psi_n(\hat{h}_{j_1}) \\ \Psi_n(\hat{h}_{j_2}) \end{pmatrix} = \frac{1}{n}\sum_{i=1}^n \psi_{\hat{\theta}}(\tilde{\boldsymbol{x}}^i) = \frac{1}{n}\sum_{i=1}^n \begin{pmatrix} \hat{\tau}^i_{j_1,j_2} - T(\hat{\sigma}_{j_1,j_2}; \{\hat{h}_{j_1}, \hat{h}_{j_2}\}) \\ \hat{\tau}^i_{j_1} - \bar{\Phi}(\hat{h}_{j_1}) \\ \hat{\tau}^i_{j_2} - \bar{\Phi}(\hat{h}_{j_2}) \end{pmatrix} = \mathbf{0}. \tag{26}
$$

$$
\Psi_n(\theta_0) = \begin{pmatrix} \Psi_n(\sigma_{j_1,j_2}) \\ \Psi_n(h_{j_1}) \\ \Psi_n(h_{j_2}) \end{pmatrix} = \frac{1}{n}\sum_{i=1}^n \psi_{\theta_0}(\tilde{\boldsymbol{x}}^i) = \frac{1}{n}\sum_{i=1}^n \begin{pmatrix} \hat{\tau}^i_{j_1,j_2} - T(\sigma_{j_1,j_2}; \{h_{j_1}, h_{j_2}\}) \\ \hat{\tau}^i_{j_1} - \bar{\Phi}(h_{j_1}) \\ \hat{\tau}^i_{j_2} - \bar{\Phi}(h_{j_2}) \end{pmatrix}. \tag{27}
$$

The difference between the estimated parameter with the true parameter can be expressed as

$$
\begin{aligned}
\hat{\theta} - \theta_0 = \begin{pmatrix} \hat{\sigma}_{j_1,j_2} - \sigma_{j_1,j_2} \\ \hat{h}_{j_1} - h_{j_1} \\ \hat{h}_{j_2} - h_{j_2} \end{pmatrix} = -\frac{1}{n}\sum_{i=1}^n & \begin{pmatrix} \frac{\partial\Psi_n(\sigma_{j_1,j_2})}{\partial\sigma_{j_1,j_2}} & \frac{\partial\Psi_n(\sigma_{j_1,j_2})}{\partial h_{j_1}} & \frac{\partial\Psi_n(\sigma_{j_1,j_2})}{\partial h_{j_2}} \\ \frac{\partial\Psi_n(h_{j_1})}{\partial\sigma_{j_1,j_2}} & \frac{\partial\Psi_n(h_{j_1})}{\partial h_{j_1}} & \frac{\partial\Psi_n(h_{j_1})}{\partial h_{j_2}} \\ \frac{\partial\Psi_n(h_{j_2})}{\partial\sigma_{j_1,j_2}} & \frac{\partial\Psi_n(h_{j_2})}{\partial h_{j_1}} & \frac{\partial\Psi_n(h_{j_2})}{\partial h_{j_2}} \end{pmatrix}^{-1} \\
& \cdot \begin{pmatrix} \hat{\tau}^i_{j_1,j_2} - T(\sigma_{j_1,j_2}; \{h_{j_1}, h_{j_2}\}) \\ \hat{\tau}^i_{j_1} - \Phi(h_{j_1}) \\ \hat{\tau}^i_{j_2} - \Phi(h_{j_2}) \end{pmatrix},
\end{aligned}
\tag{28}
$$

---

[1] We will not provide proof of this in this paper; however, interested readers may refer to [33]

where the specific form of each entry of the gradient matrix is expressed as

$$\frac{\partial \Psi_n(\sigma_{j_1,j_2})}{\partial \sigma_{j_1,j_2}} = -\frac{1}{2\pi\sqrt{(1-\sigma_{j_1,j_2}^2)}}\exp\left(-\frac{(h_{j_1}^2 - 2\sigma_{j_1,j_2}h_{j_1}h_{j_2} + h_{j_2}^2)}{2(1-\sigma_{j_1,j_2}^2)}\right);$$

$$\frac{\partial \Psi_n(\sigma_{j_1,j_2})}{\partial h_{j_1}} = \int_{h_{j_2}}^{\infty} \frac{1}{2\pi\sqrt{1-\sigma_{j_1,j_2}^2}}\exp\left(-\frac{h_{j_1}^2 - 2\sigma_{j_1,j_2}h_{j_1}x_2 + x_2^2}{2(1-\sigma_{j_1,j_2}^2)}\right)dx_2;$$

$$\frac{\partial \Psi_n(\sigma_{j_1,j_2})}{\partial h_{j_2}} = \int_{h_{j_1}}^{\infty} \frac{1}{2\pi\sqrt{1-\sigma_{j_1,j_2}^2}}\exp\left(-\frac{h_2^2 - 2\sigma_{j_1,j_2}h_{j_2}x_1 + x_1^2}{2(1-\sigma_{j_1,j_2}^2)}\right)dx_1;$$

$$\frac{\partial \Psi_n(h_{j_1})}{\partial \sigma_{j_1,j_2}} = 0;$$

$$\frac{\partial \Psi_n(h_{j_1})}{\partial h_{j_1}} = \frac{1}{\sqrt{2\pi}}\exp\left(-\frac{h_{j_1}^2}{2}\right); \tag{29}$$

$$\frac{\partial \Psi_n(h_{j_1})}{\partial h_{j_2}} = 0;$$

$$\frac{\partial \Psi_n(h_{j_2})}{\partial \sigma_{j_1,j_2}} = 0;$$

$$\frac{\partial \Psi_n(h_{j_2})}{\partial h_{j_1}} = 0;$$

$$\frac{\partial \Psi_n(h_{j_2})}{\partial h_{j_2}} = \frac{1}{\sqrt{2\pi}}\exp\left(-\frac{h_{j_2}^2}{2}\right).$$

For simplicity of notation, we define

$$\hat{\sigma}_{j_1,j_2} - \sigma_{j_1,j_2} = \frac{1}{n}\sum_{i=1}^{n}\xi_{j_1,j_2}^i, \tag{30}$$

where the specific form is of $\{\xi_{j_1,j_2}^i\}$ is defined in Eq. (28). We should note that $\{\xi_{j_1,j_2}^i\}$ are i.i.d random variables with mean zero (this property will be the key to the derivation of inference of CI). As long as our estimation $\hat{\theta}$ converge in probability to $\theta_0$ as proved in A.1, we have

$$\sqrt{n}(\hat{\theta} - \theta_0) \rightsquigarrow N\left(0, ((P\psi_{\theta_0}')^{-1}P\psi_{\theta_0}\psi_{\theta_0}^T(P\psi_{\theta_0}'^T)^{-1})_{1,1}\right), \tag{31}$$

where $\psi_{\theta_0}$ is defined in Eq. (27). However, in practice, we don't have access to either $P$ or $\theta_0$. In this scenario, we can plug in the empirical distribution of $\mathbb{P}_n\psi_{\hat{\theta}}$ to get the estimated variance, i.e., the actual variance used in the calculation of $\hat{\sigma}_{j_1,j_2} - \sigma_{j_1,j_2}$ is

$$\frac{1}{n}\left((\mathbb{P}_n\psi_{\hat{\theta}}')^{-1}\mathbb{P}_n\psi_{\hat{\theta}}\psi_{\hat{\theta}}^T(\mathbb{P}_n\psi_{\hat{\theta}}'^T)^{-1}\right)_{1,1}. \tag{32}$$

## A.5    Derivation of Lem. 2.8

Use the same line of procedure as in the derivation of Lem. 2.7, for mixed pair of observations where $X_{j_1}$ is continuous and $\tilde{X}_{j_2}$ is discrete, we can construct the criterion function

$$\Psi_n(\hat{\theta}) = \begin{pmatrix} \Psi_n(\hat{\sigma}_{j_1,j_2}) \\ \Psi_n(\hat{h}_{j_2}) \end{pmatrix} = \frac{1}{n}\sum_{i=1}^{n}\psi_{\hat{\theta}}(\tilde{\boldsymbol{x}}^i) = \frac{1}{n}\sum_{i=1}^{n}\begin{pmatrix} \hat{\tau}_{j_1,j_2}^i - T(\hat{\sigma}_{j_1,j_2}; \{0, \hat{h}_{j_2}\}) \\ \hat{\tau}_{j_2}^i - \bar{\Phi}(\hat{h}_{j_2}) \end{pmatrix} = \boldsymbol{0}. \tag{33}$$

$$\Psi_n(\theta_0) = \begin{pmatrix} \Psi_n(\sigma_{j_1,j_2}) \\ \Psi_n(h_{j_2}) \end{pmatrix} = \frac{1}{n}\sum_{i=1}^{n}\psi_{\theta_0}(\tilde{\boldsymbol{x}}^i) = \frac{1}{n}\sum_{i=1}^{n}\begin{pmatrix} \hat{\tau}_{j_1,j_2}^i - T(\sigma_{j_1,j_2}; \{0, h_{j_2}\}) \\ \hat{\tau}_{j_2}^i - \bar{\Phi}(h_{j_2}) \end{pmatrix}. \tag{34}$$

The difference between the estimated parameter with the true parameter can be expressed as

$$
\hat{\theta}-\theta_0 = \begin{pmatrix} \hat{\sigma}_{j_1,j_2} - \sigma_{j_1,j_2} \\ \hat{h}_{j_2} - h_{j_2} \end{pmatrix} = -\frac{1}{n}\sum_{i=1}^{n} \begin{pmatrix} \frac{\partial \Psi_n(\sigma_{j_1,j_2})}{\partial \sigma_{j_1,j_2}} & \frac{\partial \Psi_n(\sigma_{j_1,j_2})}{\partial h_{j_2}} \\ \frac{\partial \Psi_n(h_{j_2})}{\partial \sigma_{j_1,j_2}} & \frac{\partial \Psi_n(h_{j_2})}{\partial h_{j_2}} \end{pmatrix}^{-1} \begin{pmatrix} \hat{\tau}^i_{j_1,j_2} - T(\sigma_{j_1,j_2}; \{0, h_{j_2}\}) \\ \hat{\tau}^i_{j_2} - \bar{\Phi}(h_{j_2}). \end{pmatrix},
$$

(35)

where the specific form of each entry of the gradient matrix can be found in Eq. (29). Using exactly the same procedure, we should have the same formation of the variance calculated as Eq. (32) with a different definition of $\psi_{\theta_0}$ and $\psi_{\hat{\theta}}$ defined in Eq. (34) (33).

### A.6 Derivation of Lem. 2.6

Use the same line of procedure as in derivation of Lem. 2.7, for a continuous pair of variables, we can construct the criterion function

$$
\Psi_n(\hat{\theta}) = \Psi_n(\hat{\sigma}_{j_1,j_2}) = \frac{1}{n}\sum_{i=1}^{n} x^i_{j_1} x^i_{j_2} - \frac{1}{n}\sum_{i=1}^{n} x^i_{j_1} \frac{1}{n}\sum_{i=1}^{n} x^i_{j_2} - \hat{\sigma}_{j_1,j_2} = 0.
$$

(36)

$$
\Psi_n(\theta_0) = \Psi_n(\sigma_{j_1,j_2}) = \frac{1}{n}\sum_{i=1}^{n} x^i_{j_1} x^i_{j_2} - \frac{1}{n}\sum_{i=1}^{n} x^i_{j_1} \frac{1}{n}\sum_{i=1}^{n} x^i_{j_2} - \sigma_{j_1,j_2}.
$$

(37)

Denote $\frac{1}{n}\sum_{i=1}^{n} x^i_{j_1}$ as $\bar{x}_{j_1}$ and $\frac{1}{n}\sum_{i=1}^{n} x^i_{j_2}$ as $\bar{x}_{j_2}$. We should have

$$
\hat{\sigma}_{j_1,j_2} - \sigma_{j_1,j_2} = \frac{1}{n}\sum_{i=1}^{n} x^i_{j_1} x^i_{j_2} - \bar{x}_{j_1}\bar{x}_{j_2} - \sigma_{j_1,j_2}.
$$

(38)

According to Eq. (22), we have

$$
\sqrt{n}(\hat{\sigma}_{j_1,j_2} - \sigma_{j_1,j_2}) \rightsquigarrow N\left(0, \frac{P\psi_{\theta_0}^2}{(P\psi'_{\theta_0})^2}\right).
$$

(39)

where $(P\psi'_{\theta_0})^2 = 1$. In practical calculation, we have the variance

$$
\frac{1}{n}\mathbb{P}_n\psi_{\hat{\theta}}^2/(\mathbb{P}_n\psi'_{\hat{\theta}})^2 = \frac{1}{n^2}\sum_{i=1}^{n}(x^i_{j_1} x^i_{j_2} - \bar{x}_{j_1}\bar{x}_{j_2} - \hat{\sigma}_{j_1,j_2})^2.
$$

(40)

### A.7 Proof of Thm. 2.9

### A.7.1 Proof of Relation between $\Sigma$, $\Omega$ with $\beta$

Consider our latent continuous variables $\boldsymbol{X} = (X_1, \ldots, X_p) \sim N(0, \boldsymbol{\Sigma})$ and do nodewise regression

$$
X_j = X_{-j}\beta_j + \epsilon_j.
$$

(41)

We can divide its covariance $\boldsymbol{\Sigma}$ and its precision matrix $\Omega = \boldsymbol{\Sigma}^{-1}$ into $X$ and $Y$ part in our regression:

$$
\boldsymbol{\Sigma} = \begin{pmatrix} \boldsymbol{\Sigma}_{jj} & \boldsymbol{\Sigma}_{j-j} \\ \boldsymbol{\Sigma}_{-jj} & \boldsymbol{\Sigma}_{-j-j} \end{pmatrix} \quad \boldsymbol{\Omega} = \begin{pmatrix} \boldsymbol{\Omega}_{jj} & \boldsymbol{\Omega}_{j-j} \\ \boldsymbol{\Omega}_{-jj} & \boldsymbol{\Omega}_{-j-j} \end{pmatrix}.
$$

(42)

Just like regular linear regression, we can get

$$
n \to \infty, \quad \beta_j = \boldsymbol{\Sigma}_{-j-j}^{-1}\boldsymbol{\Sigma}_{-jj}.
$$

(43)

From the invertibility of a block matrix

$$
\begin{bmatrix} A & B \\ C & D \end{bmatrix}^{-1} = \begin{bmatrix} (A - BD^{-1}C)^{-1} & -(A - BD^{-1}C)^{-1}BD^{-1} \\ -D^{-1}C(A - BD^{-1}C)^{-1} & D^{-1} + D^{-1}C(A - BD^{-1}C)^{-1}BD^{-1} \end{bmatrix}.
$$

(44)

If $A$ and $D$ is invertible, we will have

$$
\begin{bmatrix} A & B \\ C & D \end{bmatrix}^{-1} = \begin{bmatrix} A - BD^{-1}C & 0 \\ 0 & (D - CA^{-1}B)^{-1} \end{bmatrix} \begin{bmatrix} I & -BD^{-1} \\ -CA^{-1} & I \end{bmatrix}.
$$

(45)

557 Thus, we can get:

$$\mathbf{\Omega}_{jj} = \mathbf{\Sigma}_{jj} - (\mathbf{\Sigma}_{j-j}\mathbf{\Sigma}_{-j-j}^{-1}\mathbf{\Sigma}_{-jj})^{-1};$$
$$\mathbf{\Omega}_{j-j} = -\left(\mathbf{\Sigma}_{jj} - (\mathbf{\Sigma}_{j-j}\mathbf{\Sigma}_{-j-j}^{-1}\mathbf{\Sigma}_{-jj})^{-1}\right)\mathbf{\Sigma}_{j-j}(\mathbf{\Sigma}_{-j-j})^{-1}. \tag{46}$$

558 Move one step forward:

$$-\mathbf{\Omega}_{jj}^{-1}\mathbf{\Omega}_{j-j} = \mathbf{\Sigma}_{j-j}(\mathbf{\Sigma}_{-j-j})^{-1}. \tag{47}$$

559 Take transpose for both sides, as long as $\mathbf{\Omega}$ is a symmetric matrix and $\mathbf{\Omega}_{-jj} = \mathbf{\Omega}_{j-j}^{T}$, we will have

$$-\mathbf{\Omega}_{jj}^{-1}\mathbf{\Omega}_{-jj} = \mathbf{\Sigma}_{-j-j}^{-1}\mathbf{\Sigma}_{-jj} = \beta_j. \tag{48}$$

560 We should note testing $\mathbf{\Omega}_{-jj} = 0$ is equivalent to testing $\beta_j = 0$ as the $\mathbf{\Omega}_{jj}$ will always be nonzero.
561 The variable $\mathbf{\Omega}_{-jj}$ captures the CI of $X_j$ with other variables. As long as the variable $\mathbf{\Omega}_{jj}$ is just one
562 scalar, we can get

$$\beta_{j,k} = -\frac{\omega_{j,k}}{\omega_{j,j}} \tag{49}$$

563 capturing the independence relationship between variable $X_j$ with $X_k$ conditioning on all other
564 variables.

### A.7.2 Detailed derivation of inference for $\beta_j$

566 Nodewise regression allows us to use the regression parameter $\beta_j$ as the surrogate of $\Omega_{-jj}$. The
567 problem now transfers to constructing the inference for $\beta_j$, specifically, the derivation of distribution
568 of $\hat{\beta}_j - \beta_j$. The overarching concept is that we are already aware of the distribution of $\hat{\sigma}_{j_1,j_2} - \sigma_{j_1,j_2}$
569 and we know that there exists a deterministic relationship between $\beta_j$ with $\mathbf{\Sigma}$. Consequently, we can
570 express $\hat{\beta}_j - \beta_j$ as a composite of $\hat{\sigma}_{j_1,j_2} - \sigma_{j_1,j_2}$ to establish such an inference. Specifically, we have

$$\begin{aligned}
\hat{\beta}_j - \beta_j &= \hat{\mathbf{\Sigma}}_{-j-j}^{-1}\hat{\mathbf{\Sigma}}_{-jj} - \mathbf{\Sigma}_{-j-j}^{-1}\mathbf{\Sigma}_{-jj} \\
&= \hat{\mathbf{\Sigma}}_{-j-j}^{-1}\left(\hat{\mathbf{\Sigma}}_{-jj} - \hat{\mathbf{\Sigma}}_{-j-j}\mathbf{\Sigma}_{-j-j}^{-1}\mathbf{\Sigma}_{-jj}\right) \\
&= -\hat{\mathbf{\Sigma}}_{-j-j}^{-1}\left(\hat{\mathbf{\Sigma}}_{-j-j}\beta_j - \mathbf{\Sigma}_{-j-j}\beta_j + \mathbf{\Sigma}_{-j-j}\beta_j - \hat{\mathbf{\Sigma}}_{-jj}\right) \\
&= -\hat{\mathbf{\Sigma}}_{-j-j}^{-1}\left((\hat{\mathbf{\Sigma}}_{-j-j} - \mathbf{\Sigma}_{-j-j})\beta_j - (\hat{\mathbf{\Sigma}}_{-jj} - \mathbf{\Sigma}_{-jj})\right),
\end{aligned} \tag{50}$$

571 where each entry in matrix $(\hat{\mathbf{\Sigma}}_{-j-j} - \mathbf{\Sigma}_{-j-j})$ and $(\hat{\mathbf{\Sigma}}_{-jj} - \mathbf{\Sigma}_{-jj})$ denotes the difference between
572 estimated covariance with true covariance. Suppose that we want to test the CI of the variable $X_1$
573 with other variables, $j = 1$, then

$$\hat{\mathbf{\Sigma}}_{-j-j} - \mathbf{\Sigma}_{-j-j} = \begin{bmatrix} \hat{\sigma}_{1,1}\dots\hat{\sigma}_{1,j-1},\hat{\sigma}_{1,j+1}\dots\hat{\sigma}_{1,p} \\ \dots \\ \hat{\sigma}_{j-1,1}\dots\hat{\sigma}_{j-1,j-1},\hat{\sigma}_{j-1,j+1}\dots\hat{\sigma}_{j-1,p} \\ \dots \\ \hat{\sigma}_{p,1}\dots\hat{\sigma}_{p,j-1},\hat{\sigma}_{p,j+1}\dots\hat{\sigma}_{p,p} \end{bmatrix} \tag{51}$$

$$-\begin{bmatrix} \sigma_{1,1}\dots\sigma_{1,j-1},\sigma_{1,j+1}\dots\sigma_{1,p} \\ \dots \\ \sigma_{j-1,1}\dots\sigma_{j-1,j-1},\sigma_{j-1,j+1}\dots\sigma_{j-1,p} \\ \dots \\ \sigma_{p,1}\dots\sigma_{p,j-1},\sigma_{p,j+1}\dots\sigma_{p,p}. \end{bmatrix}. \tag{52}$$

574 Suppose that we want to test the CI of the variable $X_1$ with other variables, $j = 1$. then

$$\hat{\mathbf{\Sigma}}_{-1-1} - \mathbf{\Sigma}_{-1-1} = \begin{bmatrix} \hat{\sigma}_{2,2}\dots\hat{\sigma}_{2,p} \\ \dots \\ \hat{\sigma}_{p,2}\dots\hat{\sigma}_{p,p} \end{bmatrix} - \begin{bmatrix} \sigma_{2,2}\dots\sigma_{2,p} \\ \dots \\ \sigma_{p,2}\dots\sigma_{p,p} \end{bmatrix} \tag{53}$$

$$:= \frac{1}{n}\sum_{i=1}^{n} \begin{bmatrix} \xi_{2,2}^i\dots\xi_{2,p}^i \\ \dots \\ \xi_{p,2}^i\dots\xi_{p,p}^i \end{bmatrix}, \tag{54}$$

where $\{\xi^i_{j_1,j_2}\}$ are i.i.d random variables with specific form defined in Eq. (28) for discrete case, Eq. (35) for mixed case and Eq. (38) in continuous case. Put them together:

$$
\begin{bmatrix} \hat{\beta}_{1,2} - \beta_{1,2} \\ \hat{\beta}_{1,3} - \beta_{1,3} \\ \cdots \\ \hat{\beta}_{1,p} - \beta_{1,p} \end{bmatrix} = -\hat{\Sigma}^{-1}_{-1-1} \frac{1}{n} \sum_{i=1}^{n} \left( \begin{bmatrix} \xi^i_{2,2} & \xi^i_{2,3} & \cdots & \xi^i_{2,p} \\ \xi^i_{3,2} & \xi^i_{3,3} & \cdots & \xi^i_{3,p} \\ \cdots & \cdots & \cdots & \cdots \\ \xi^i_{p,2} & \xi^i_{p,3} & \cdots & \xi^i_{p,p} \end{bmatrix} \begin{bmatrix} \beta_{1,2} \\ \beta_{1,3} \\ \cdots \\ \beta_{1,p} \end{bmatrix} - \begin{bmatrix} \xi^i_{2,1} \\ \xi^i_{3,1} \\ \cdots \\ \xi^i_{p,1} \end{bmatrix} \right). \tag{55}
$$

As $\frac{1}{n} \sum_{i=1}^{n} \xi^i_{j_1,j_2}$ is asymptotically normal, the who vector of $\hat{\beta}_1 - \beta_1$ is a linear combination of Gaussian distribution. However, We cannot merely engage in a linear combination of its variance as they are dependent with each other. For example, if $Y_1, Y_2$ are dependent and we are trying to find out $Var(aY_1 + bY_2)$, we should have

$$
Var(aY_1 + bY_2) = \begin{bmatrix} a & b \end{bmatrix} \begin{bmatrix} Var(Y_1) & Cov(Y_1, Y_2) \\ Cov(Y_1, Y_2) & Var(Y_2) \end{bmatrix} \begin{bmatrix} a \\ b \end{bmatrix}. \tag{56}
$$

Now, suppose we are interested in the distribution of $\hat{\beta}_{1,2} - \beta_{1,2}$, we should have

$$
\hat{\beta}_{1,2} - \beta_{1,2} = \frac{1}{n} \sum_{i=1}^{n} (\hat{\Sigma}^{-1}_{-1-1})_{[2],:} \left( \begin{bmatrix} \xi^i_{2,2} & \xi^i_{2,3} & \cdots & \xi^i_{2,p} \\ \xi^i_{3,2} & \xi^i_{3,3} & \cdots & \xi^i_{3,p} \\ \cdots & \cdots & \cdots & \cdots \\ \xi^i_{p,2} & \xi^i_{p,3} & \cdots & \xi^i_{p,p} \end{bmatrix} \begin{bmatrix} \beta_{1,2} \\ \beta_{1,3} \\ \cdots \\ \beta_{1,p} \end{bmatrix} - \begin{bmatrix} \xi^i_{2,1} \\ \xi^i_{3,1} \\ \cdots \\ \xi^i_{p,1} \end{bmatrix} \right), \tag{57}
$$

where $(\hat{\Sigma}^{-1}_{-1-1})_{[2],:}$ is the row of index of $X_2$ of $\hat{\Sigma}^{-1}_{-1-1}$ ([2] denotes the index of the variable). For ease of notation, let

$$
\Xi^i_{-1,-1} = \begin{bmatrix} \xi^i_{2,2} & \xi^i_{2,3} & \cdots & \xi^i_{2,p} \\ \xi^i_{3,2} & \xi^i_{3,3} & \cdots & \xi^i_{3,p} \\ \cdots & \cdots & \cdots & \cdots \\ \xi^i_{p,2} & \xi^i_{p,3} & \cdots & \xi^i_{p,p} \end{bmatrix}, \qquad \Xi^i_{-1,1} = \begin{bmatrix} \xi^i_{2,1} \\ \xi^i_{3,1} \\ \cdots \\ \xi^i_{p,1} \end{bmatrix}, \tag{58}
$$

and let

$$
B^i_{-1} = \begin{pmatrix} \xi^i_{2,1} & \xi^i_{3,1} & \cdots & \xi^i_{p,1} \\ \xi^i_{2,2} & \xi^i_{2,3} & \cdots & \xi^i_{2,p} \\ \xi^i_{3,2} & \xi^i_{3,3} & \cdots & \xi^i_{3,p} \\ \cdots & \cdots & \cdots & \cdots \\ \xi^i_{p,2} & \xi^i_{p,3} & \cdots & \xi^i_{p,p} \end{pmatrix} \tag{59}
$$

as the concatenation of those two matrices. The variance is calculated as

$$
Var\left( \sqrt{n}(\hat{\beta}_{1,2} - \beta_{1,2}) \right) = a^{[2]T} \frac{1}{n} \sum_{i=1}^{n} vec(B^i_{-1}) vec(B^i_{-1})^T a^{[2]}, \tag{60}
$$

where

$$
a^{[2]}_l = \begin{cases} \left( \hat{\Sigma}^{-1}_{-1-1} \right)_{[2],l}, & \text{for } l \in \{1, \ldots, p-1\} \\ \sum_{q=1}^{n} \left( \hat{\Sigma}^{-1}_{-1-1} \right)_{[2],l} (\beta_1)_q, & \text{for } l \in \{p, \ldots, p^2 - p\} \end{cases} \tag{61}
$$

$vec(B^i_{-1})$ is the squeezed vector form of matrix $vec(B^i_{-1}) \in \mathbb{R}^{p \times p-1}$, i.e.,

$$
vec(B^i_{-1}) = \begin{pmatrix} \xi^i_{2,1} \\ \xi^i_{3,1} \\ \vdots \\ \xi^i_{p,p} \end{pmatrix}. \tag{62}
$$

Thus, the distribution of $\hat{\beta}_{j,k} - \beta_{j,k}$ is

$$
\hat{\beta}_{j,k} - \beta_{j,k} \sim N(0, a^{[k]T} \frac{1}{n^2} \sum_{i=1}^{n} vec(B^i_{-j}) vec(B^i_{-j})^T a^{[k]}). \tag{63}
$$

In practice, we can plug in the estimates of $\beta_j$ to estimate the interested distribution and do the CI test by hypothesizing $\beta_{j,k} = 0$.

## A.8 Discussion of assumption of zero mean and identity variance

In this section, we engage in a more thorough discussion regarding our assumptions about $\boldsymbol{X}$. Specifically, we demonstrate that this assumption of mean and variance does not compromise the generality. In other words, the true model may possess different mean and variance values, but we proceed by treating it as having a mean of zero and identity variance.

The key ingredient allowing us to assume such a model is, the discretization function $g_j$ is an unknown nonlinear monotonic function. Suppose the $g'_j$ maps the continuous domain to a binary variable, and we have the "groundtruth" variable, denoted $X'_j$, with mean $a$ and variance $b$. Assume the cardinality of the discretized domain is only 2, i.e., our observation $\tilde{X}_j$ can only be 0 or 1. We further have the constant $d'_j$ as the discretization boundary such that we have the observation

$$\tilde{X}_j = \mathbb{1}(g'_j(X'_j) > d'_j) = \mathbb{1}(X'_j > g'^{-1}_j(d_j))$$

We can always produce our assumed variable $X_j$ with mean 0 and variance 1, such that $X_j = \frac{1}{\sqrt{b}}X'_j - \frac{a}{\sqrt{b}}$ and the same observation with a different nonlinear transformation $g_j$ and decision boundary $d_j$, such that

$$\tilde{X}_j = \mathbb{1}(g_j(X_j) > d_j) = \mathbf{1}(X_j > g^{-1}_j(d_j)) = \mathbb{1}(X'_j > \sqrt{b}g^{-1}_j(d_j) + a)$$

As long as the observation $\tilde{X}_j$ is the same, we should have $\sqrt{b}g^{-1}_j(d_j) + a = g'^{-1}_j(d_j)$. Our assumed model $X_j$ clearly mimics the "groundtruth" $X'_j$. Besides, according to Lem. A.2, we have one-to-one mapping between $\hat{\tau}_{j_1 j_2}$ with the estimated covariance for fixed $\hat{h}_{j_1}, \hat{h}_{j_2}$. Thus, as long as the observation is the same, the estimation of covariance $\hat{\sigma}_{j_1, j_2}$ remains unaffected by our assumptions regarding the mean and variance of $\boldsymbol{X}$, so do the following inference.

We further conduct casual discovery experiments to empirically validate our statement, which is shown in App. C.3.

## B   Data Generation and Figure of main experiments: causal discovery

**Data Generation and Code**   We construct the true DAG $\mathcal{G}$ using the Bipartite Pairing (BP) model [2], with the number of edges being one fewer than the number of nodes. The subsequent generation of true multivariate Gaussian data involves assigning causal weights drawn from a uniform distribution $U \sim (0.5, 2)$ and incorporating noise via samples from a standard normal distribution for each variable. Following this, we binarize the data, setting the threshold randomly based on each variable's range. The code implementation is based on [40] .

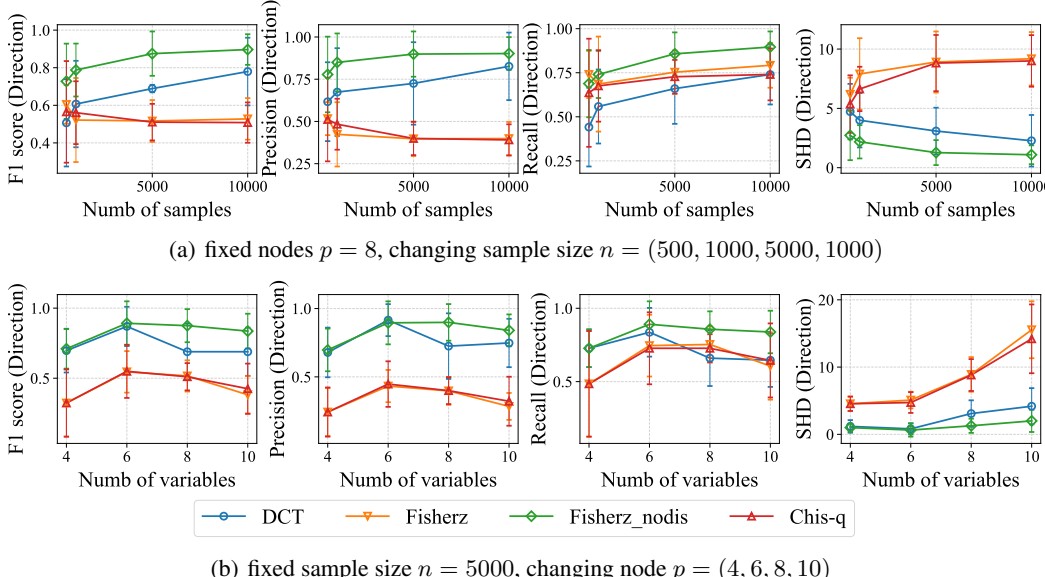

(a) fixed nodes $p = 8$, changing sample size $n = (500, 1000, 5000, 1000)$

(b) fixed sample size $n = 5000$, changing node $p = (4, 6, 8, 10)$

Figure 4: Experiment result of DAG discovery on synthetic data for changing sample size (a) and changing number of nodes (b). Fisherz_nodis is the Fisher-z test applied to original continuous data. We evaluate $F_1$ (↑), Precision (↑), Recall (↑) and SHD (↓).

## C  Additional experiments

### C.1  Linear non-Gaussian and nonlinear

Our model requires that the original data must adhere to the hypothesis of following a multivariate normal distribution, which appears to potentially limit the generalizability. Therefore, it is worthwhile to explore its robustness when such assumptions are violated. In this regard, we conducted several experiments, including scenarios involving linear non-Gaussian and nonlinear Gaussian.

For both cases, we follow the setting of our experiment where there are $p = 8$ nodes and $p - 1$ edges. We explore the effect of changing sample size $n = (100, 500, 2000, 5000)$. Specifically for linear non-Gaussian case, we adhere to some of the settings outlined by [28], conducting experiments where the original continuous data followed: (1) a Student's t-distribution with 3 degrees of freedom, (2) a uniform distribution, and (3) an exponential distribution. Each variable is generated as $X_i = f(PA_i) + noise$, where $noise$ follows the distribution in (1), (2), (3) correspondingly and $f$ is a linear function. The first three rows of Fig. 5 and Fig. 6 show the result of the linear non-Gaussian case.

For the nonlinear cases, we follow setting in [19], where every variable $X_i$ is generated as $X_i = f(WPA_i + noise)$, $noise \sim N(0, 1)$ and $f$ is a function randomly chosen from (a) $f(x) = sin(x)$, (b) $f(x) = x^3$, (c) $f(x) = tanh(x)$, and (d) $f(x) = ReLU(x)$. $W$ is a linear function. Similarly, we set the number of nodes at $p = 8$ and change the number of samples $n = (500, 2000, 5000)$. For both cases, we run 10 graph instances with different seeds and report the result of skeleton discovery in Fig. 5 and DAG in Fig. 6 (The same orientation rules [11] used in the main experiment are employed to convert a CPDAG [6] into a DAG). The last row of Fig. 5 and Fig. 6 shows the result of the nonlinear case.

Based on the experimental outcomes, DCT demonstrates marginally superior or comparable efficacy in terms of the F1-score, precision, and SHD relative to both the Fisher-Z test and the Chi-square test when dealing with small sample sizes. Nevertheless, as the sample size increases, DCT's performance clearly surpasses that of the aforementioned tests across all three evaluated metrics, especially in the linear case. Consistent with observations from the main experiment, DCT exhibits a lower recall in comparison to the baseline tests. This discrepancy can be attributed to the baseline tests being prone to incorrectly infer conditional dependence and connect a large proportion of nodes. According to the results, our test shows notable robustness under the case assumptions are violated, confirming its practical effectiveness.

### C.2  Denser graph

DCT primarily works on cases where CI is mistakenly judged as conditional dependence due to discretization. Consequently, its efficacy is more pronounced in scenarios characterized by a relatively sparse graph, as numerous instances are truly conditionally independent. Nevertheless, the investigation of causal discovery with a dense latent graph is essential for evaluating the power of a test, i.e., its ability to successfully reject the null hypothesis when the tested pairs are conditionally dependent. Thus, we conduct the experiment where $p = 8, n = 10000$ and changing edges ($p + 2, p + 4, p + 6$). Similarly, the latent continuous data follows a multivariate Gaussian model and the true DAG $\mathcal{G}$ is constructed using BP model. We run 10 graph instances with different seeds and report the result of the skeleton discovery and DAG in Fig. 7.

According to the experiment results, DCT exhibits better performance in terms of the F1-score, precision, and SHD relative to both the Fisher-Z test and the Chi-square test. As the graph becomes progressively denser, the superiority of the Discrete Causality Test (DCT) correspondingly diminishes as there are few conditional independent cases in the true DAG. Due to the same reason, The recall remains lower than that of other baseline methods.

### C.3  multivariate Gaussian with nonzero mean and non-unit variance

We employed a setting nearly identical to the main experiment, with the only difference being the alteration in data generation: instead of using a standard normal distribution, we used a Gaussian distribution with mean sampled from $U(-2, 2)$ and variance sampled from $U(0, 3)$. We fix the number of variables as $p = 8$ and change the number of samples $n = (100, 500, 2000, 5000)$. The Fig. 8 shows the result and demonstrates the effectiveness of our method.

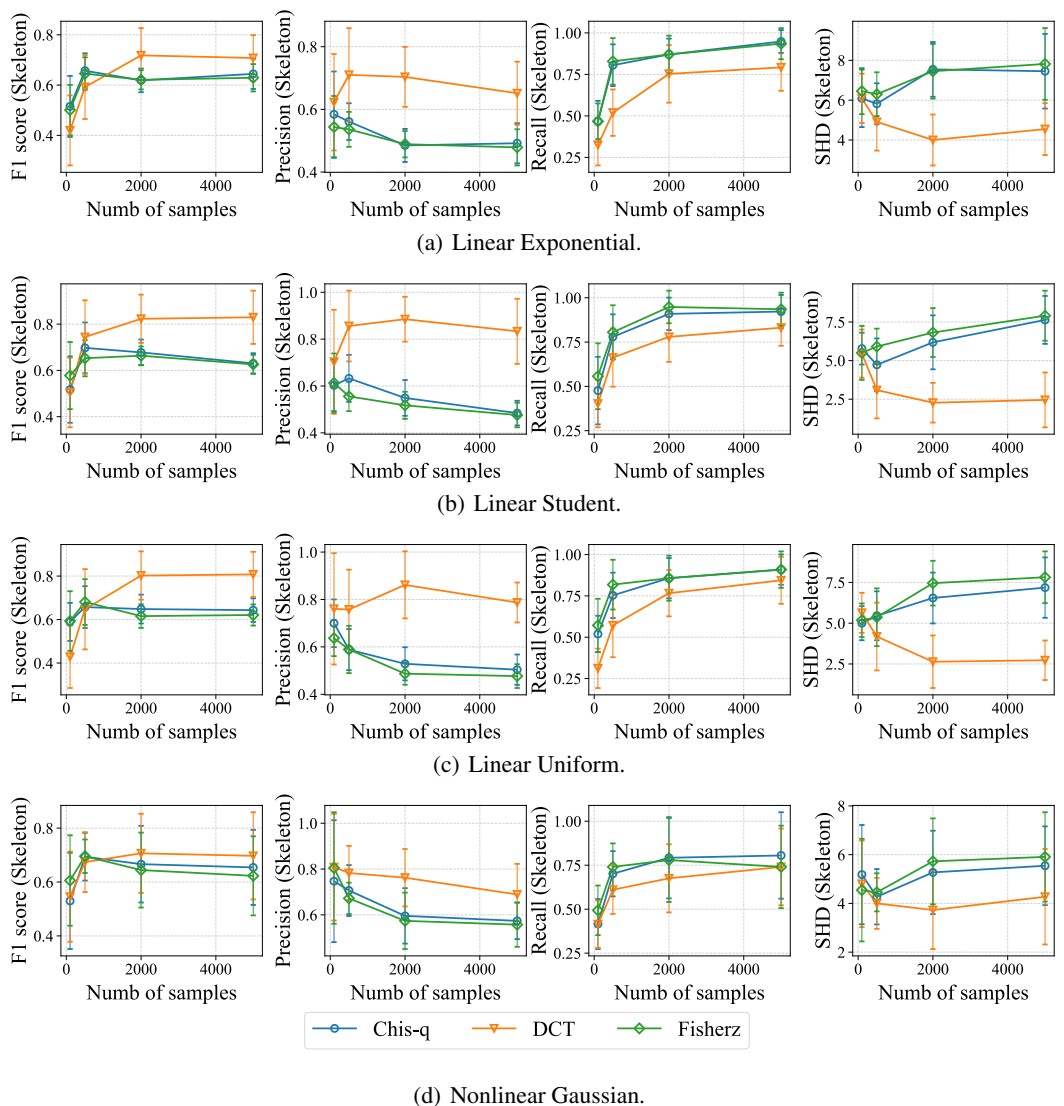

Figure 5: Experiment result of causal discovery on synthetic data with $p = 8$, $n = (100, 500, 2000, 5000)$ where the data generation process violates our assumptions. The data are generated with either nongaussian distributed (a), (b), (c) or the relations are not linear (d). The figure reports $F_1$ ($\uparrow$), Precision ($\uparrow$), Recall ($\uparrow$) and SHD ($\downarrow$) on skeleton.

## C.4 Real-world dataset

To further validate DCT, we employ it on a real-world dataset: Big Five Personality https://openpsychometrics.org/, which includes 50 personality indicators and over 19000 data samples. Each variable contains 5 possible discrete values to represent the scale of the corresponding questions, where 1=Disagree, 2=Weakly disagree, 3=Neutral, 4=Weakly agree and 5=Agree, e.g., "N3=1" means "I agree that I worry about things". This scenario clearly suits DCT, where the degree of agreement with a certain question must be a continuous variable while we can only observe the result after categorization. We choose three variables respectively: [N3: I worry about things], [N10: I often feel blue ], [N4: I seldom feel blue]. We then do the casual discovery using PC algorithm with DCT and compare it with the Chi-square test and Fisher-Z test. The result can be found in Fig. 9.

Based on the experimental outcomes, despite the absence of a groundtruth for reference, we observe that the results obtained via DCT appear more plausible than those derived from Fisher-Z and Chi-square tests. Specifically, DCT suggests the relationship $N_3 \perp\!\!\!\perp N4|N10$, which is reasonable as intuitively, the answer of 'I often feel blue' already captures the information of 'I seldom feel blue'.

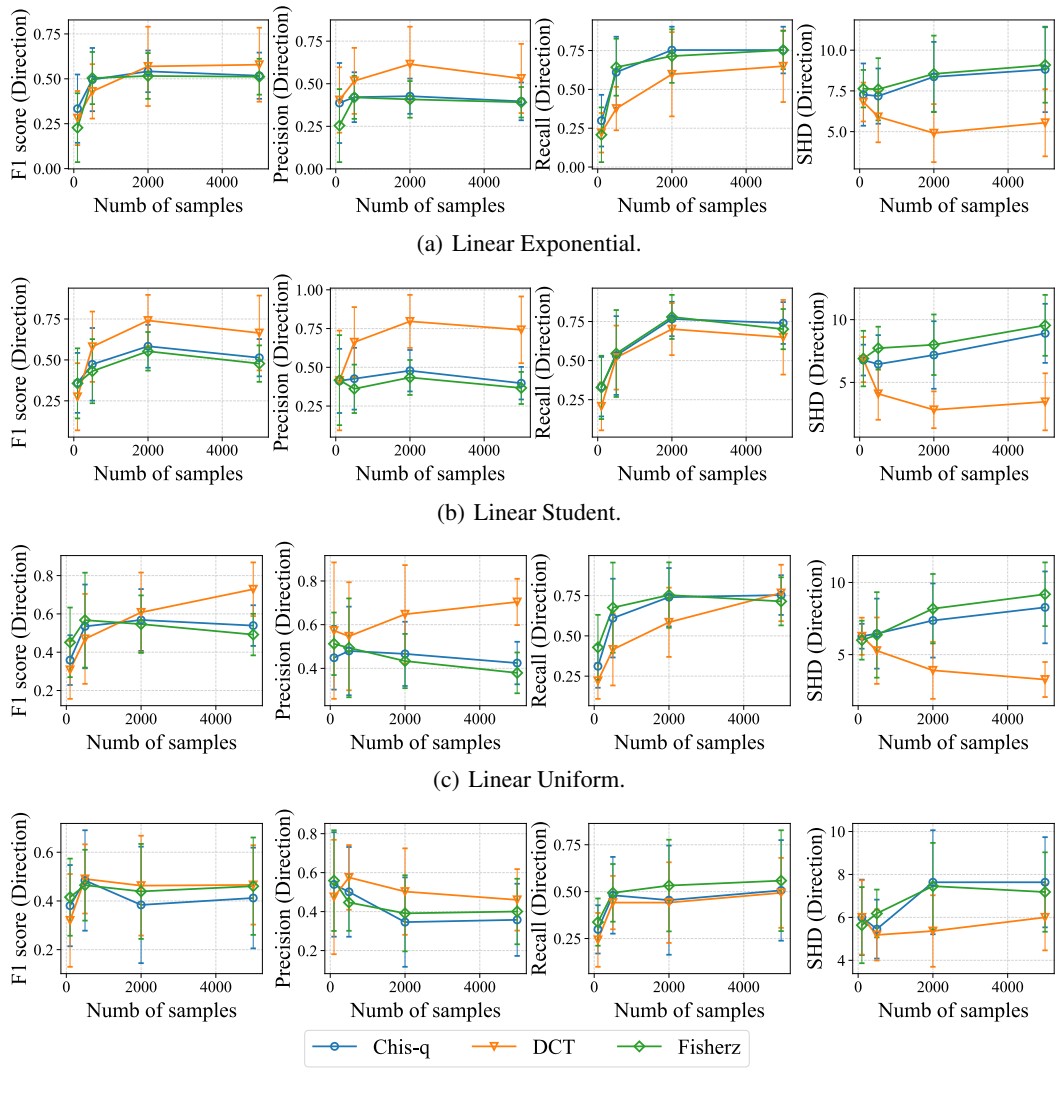

(a) Linear Exponential.

(b) Linear Student.

(c) Linear Uniform.

(d) Nonlinear Gaussian.

Figure 6: Experiment result of causal discovery on synthetic data with $p = 8$, $n = (100, 500, 2000, 5000)$ where the data generation process violates our assumptions. The data are generated with either nongaussian distributed (a), (b), (c) or the relations are not linear (d). The figure reports $F_1$ ($\uparrow$), Precision ($\uparrow$), Recall ($\uparrow$) and SHD ($\downarrow$) on DAG.

As a comparison, both Fisher-Z and Chi-square return a fully connected graph. The results directly correspond to our illustrative example shown in Fig. 1, substantiating the necessity of our proposed test.

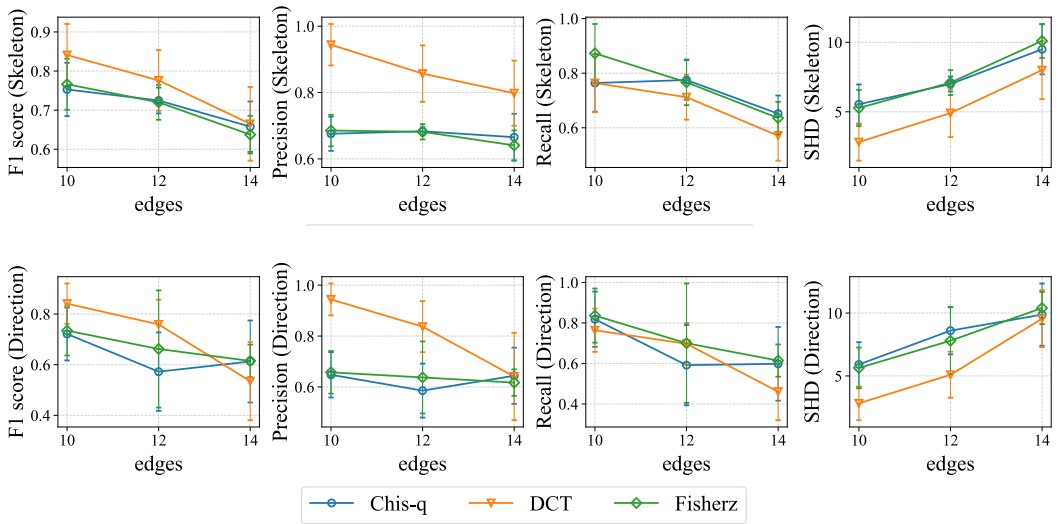

Figure 7: Experimental comparison of causal discovery on synthetic datasets for denser graphs with $p = 8, n = 10000$ and edges varying $p + 2, p + 4, p + 6$. We evaluate $F_1$ ($\uparrow$), Precision ($\uparrow$), Recall ($\uparrow$) and SHD ($\downarrow$) on both skeleton and DAG.

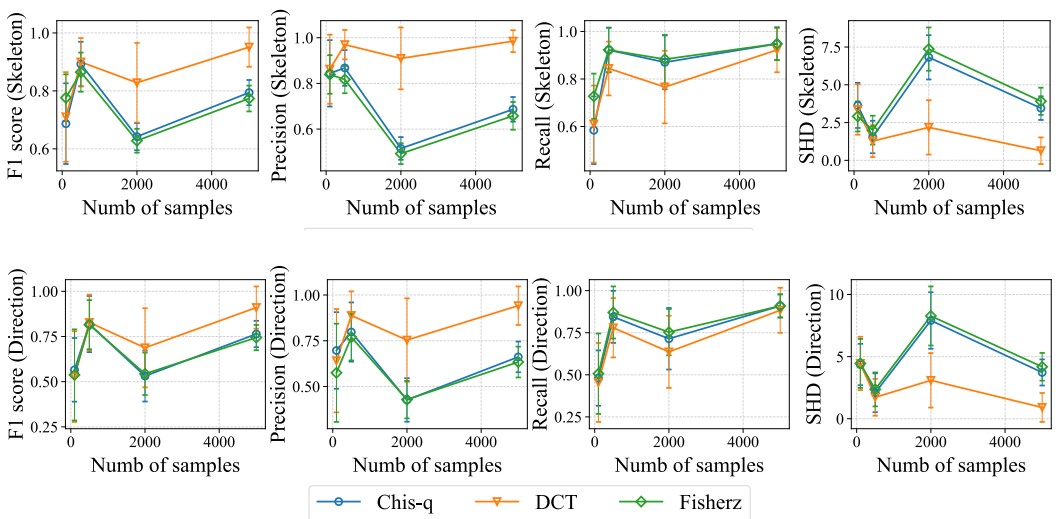

Figure 8: Experimental comparison of causal discovery on synthetic datasets for multivariate Gaussian model with $p = 8, n = (100, 500, 2000, 5000)$ and where mean is not zero. We evaluate $F_1$ ($\uparrow$), Precision ($\uparrow$), Recall ($\uparrow$) and SHD ($\downarrow$) on both skeleton and DAG.

## D Related Work

Testing for CI is pivotal in the field of causal discovery [30], and a variety of methods exist for performing CI tests (CI tests). An important group of CI test methods involves the assumption of Gaussian variables with linear dependencies. For example, under this assumption, Gaussian graphical models are extensively studied [37, 25, 22, 26]. To address CI test under Gaussian assumption, partial correlation serves as a viable method for CI testing [4]. To evaluate the independence of variables $X_1$ and $X_2$ conditional on $\boldsymbol{Z}$, The technique proposed by [32] determines CI by comparing the estimations of $p(X_1|X_2, \boldsymbol{Z})$ and $p(X_1|X_2)$.

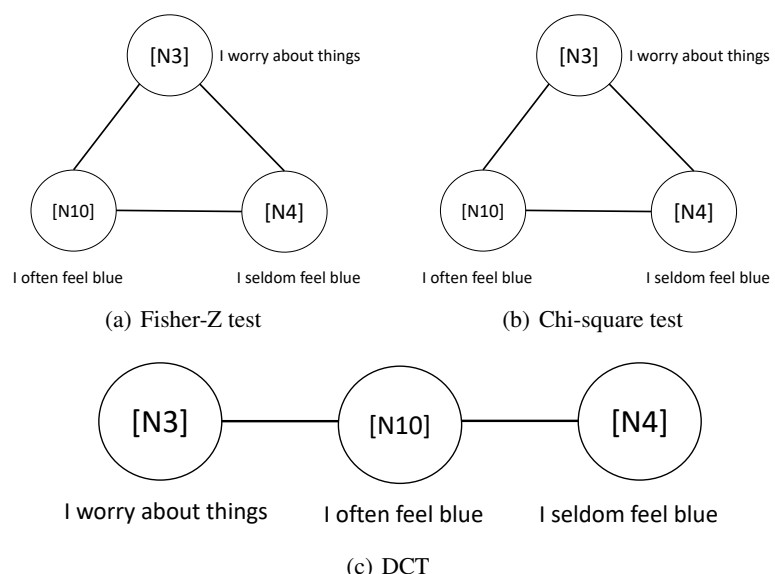

Figure 9: Experimental comparison of causal discovery on the real-world dataset.

Another approach involves discretizing $Z$ and performing independent tests within each resulting bin [21]. Our work, however, diverges from these existing methods in two significant ways. Firstly, we are equipped to handle data, where partial variables are discretized. Additionally, we postulate that discrete variables are derived from the transformation of continuous variables in a latent Gaussian model. With the same assumption, the most closely related study is by [13], where the authors developed a novel rank-based estimator for the precision matrix of mixed data. However, their work stops short of providing a CI test for this method. Our research fills this gap, offering the ability to estimate the precision matrix for both discrete and mixed data and providing a rigorous CI test for our methodology.

Recent advancements in CI testing have utilized kernel methods for continuous variables influenced by nonlinear relationships. [16] describes non-parametric CI relationships using covariance operators in reproducing kernel Hilbert spaces (RKHS). KCI test [38] assesses the partial associations of regression functions linking $x$, $y$, and $z$, while RCI test [31] aims to enhance the KCI test's efficiency. In KCIP test [12] employs permutations of samples to emulate CI scenarios. CCI test [27] further reformulates testing into a process that leverages the capabilities of supervised learning models. For discrete variable analysis, the $G^2$ test [1] and conditional mutual information [39] are commonly employed. However, their method cannot deal with our setting where only discretized version of latent variables can be observed.

# E   Resource Usage

All the experiments are run using Intel(R) Xeon(R) CPU E5-2680 v4 with 55 processors. It costs 4 hours to run experiments in Section 3.1.

# F   Limiation and Broader Impacts

**Limitation**   So far, the largest limitation of our method is to treat discretized variables as binary, which wastes the available information. Besides that, the parametric assumption limits its generalizability. However, we need to point out this is pretty normal in CI test fields.

**Broader Impacts**   The goal of our proposed method is to test the conditional independence relationship given discretized observation. This task is essential and has broad applications. We are confident that our method will be beneficial and will not result in negative societal impacts.

