# OpenReview forum: "A Conditional Independence Test in the Presence of Discretization"
_NeurIPS.cc/2024/Conference — Submitted to NeurIPS 2024_

### Official Review · Reviewer_dVXS · 2024-07-05

**Soundness:** 2
**Presentation:** 2
**Contribution:** 1
**Rating:** 3
**Confidence:** 3

**Summary:**

The authors proposes a method for testing conditional independence in presence of discretisation.
They assume the variables to be jointly Gaussian, and that some of them are accessible only after discretisation; thus the data contain a mix of discrete and continuous variables.
Discretization might remove some conditional independencies. Assume X1 to be independent from X2 given X3.
It might be that X1 becomes instead dependent on X2 given \tilde{X3}, where \tilde{X3} is the discretised  X3.
The authors develop a way to infer the latent correlation on the real value variables and they propose a novel test for conditional independence for  the setting mixed continuous and discrete variables.

**Strengths:**

Testing conditional independence with mixed types of variables is an interesting topic and the work is original.
The presentation is good, even though I could not follow the development of the bridge equations (this might be because I am not familiar with the adopted techniques).

**Weaknesses:**

* I am skeptical about the specific research question addressed.
X1 and X3 are independent given X2; yet they might be not independent given the discretised version of X2.

For instance, with ref to  Fig 1a,   X1 and X3 are formally dependent given \tilde{X2}; yet the induced dependence might be very weak.
I argue that the strength of the induced dependence depends on how the discretisation is done.
Example: X2 is human height, discretized into bins of few centimeters; then \tilde{X2}  is practically as informative as X2 and the induced dependence is likely to be negligible, in which case it might be sensible not rejecting H0.

The author did not discuss the impact  of the adopted discretization approach on the induced dependence.
Also, there is no compelling example in which discretisation induces a strong dependence.


* In the first set of experiments the test is better calibrated than the competitors, but it has by far less power. Overall, these results are not very strong.


* The competitor tests (Z-test and chi-square) are not modern.
There is no comparison against  existing tests for mixed variables; I can cite for instance Bayesian Independence Test with Mixed-type Variables, Benavoli et al. 2021.
Another simple baseline which I think should be present: test conditional independence having discretized all variables and use modern test for discrete variables (as a starting point, I suggest  those available in bnlearn https://www.bnlearn.com/documentation/man/conditional.independence.tests.html )

**Questions:**

* If we can only observe \tilde{X2},  I do not really see why we shall test conditional independence given X2.

*  The authors discretize the  data  K = (2, 4, 8, 12) levels, with boundaries randomly set based on the variable range.
Why  random boundaries? Why not to use uniformly spaced bin or equiprobable bins?
I think the specific discretization you adopt has an impact on the strength of the induced dependence.
Might it be that by discretising at random you make \tilde{X2} less informative than it could be?

**Limitations:**

No potential negative societal impact.

---

> ### Author Rebuttal · Authors · 2024-08-07
>
> >Q1:  I am skeptical about the specific research question addressed. X1 and X3 are independent given X2; yet they might be not independent given the discretised version of X2.
>
> A1: Thanks for raising the confusion and we would like to use this opportunity to state our motivation. Just as illustrated in Figure~1, directly taking some variables that are inherently continuous as discrete might lead to wrong conclusion of conditional independence. However, this situation is very common in the real world. In most scenarios, accurately measuring the exact value is not possible. Among various cases, we can only obtain a rough categorization. For example, we cannot get an exact number to reflect the severity of cancer; we only get an approximation like "phase one" to indicate its severity. Similarly, in the real-world data set we include in the paper, in psychological tests, we cannot get extremely precise numbers to reflect the respondent's agreement with a particular question; we only get a rating from 1 to 5, etc. All of those motivate us to develop a test serving as a correction, trying to infer the correct conditional independence relationship.
>
> >Q2: The strength of the induced dependence depends on how the discretization is done.
>
> A2: Thanks for your question.  We totally agree that the induced dependence is related to discretization mechanism, and with the increment of the cardinality of discretization, the induced dependence decreases. In the extreme case, when the discretization mechanism is an identity mapping, there is no information loss at all and the observed variables have exactly the same conditional independence relationship as latent variables.
>
> As mentioned in the introduction, our goal is to infer the conditional independence of variables that cannot be accurately measured, such as cancer severity and mental health status. These variables have cardinalities that we cannot control. When the cardinality is relatively small, the Type I error of traditional tests can become unacceptably high, as demonstrated in Experiment~3.1. Therefore, we propose the DCT method to specifically handle this scenario. For variables that can be accurately measured, such as height, using DCT is unnecessary.
>
> >Q3: small power
>
> A3: We have also observed that the power of the DCT in experiments can be relatively low compared with a part of baselines when the **sample size is small**. However, we must point out that this is the result **without calibration**.
>
> Just as the cases illustrated in Figure~1, the tested pairs are conditional dependent without given the discretized variables in principle. This leads to the p-value obtained from conventional tests is always close to 0. In such cases, directly comparing the obtained p-value with the preset significance level ($\alpha=0.05$) is unfair for DCT. Therefore, we provide the results of the test with **calibration** applied to cases with a relatively small sample size ($n=100,500$)  in Figure1 in the attached pdf. Specifically, we empirically determine the value corresponding to the five percent quantile under the $H_0$. We then use this value as the threshold to determine the rejection of $H_0$ in the evaluation of Type II error for a fair comparison.
>
> From the experiment result, DCT exhibits superior performance compared with Chi-square tests applied to every kinds of discrete data and Fisher-z test applied to binary data.  As the cardinality of discretization increases, the performance of DCT does not match that of Fisher-z test. This result is not surprising as the discretization drastically reduces the available information. Additionally, we need to note that all other baselines maintain significantly higher Type I error rates as shown in Figure~2 of the main text. At the same time, when the sample size increases,  Type II error dramatically decreases while maintaining ideal Type I error, demonstrating the efficacy of the proposed approach. However, for the baselines, although the Type II error is further reduced, the Type I error increases significantly. This highlights the necessity of developing tests that correctly infer conditional independence, as achieved in this paper.
>
> >Q4: other competitor tests specifically designed for discrete data
>
> A4: Thanks for your great advice. We would also use this opportunity to emphasize the fundamental difference between DCT with other tests for discrete data. **We care about the conditional independence of latent continuous variables rather than those observed discrete variables**.  Those discrete variables, which should be inherently continuous, don't reflect the real conditional independence we care about. Applying any traditional tests is highly likely to conclude conditional dependence (They are always d-connected). Larger sample size, more accurate test, higher Type I error.
>
> Following your suggestion, we compared the Type I and Type II errors on equiprobable discretized data with two new baselines: the G-square test [1] and the mutual information-based test [2]. The results, shown in Figure 2 of the attached PDF, demonstrate that DCT has more consistent Type I error control across all scenarios. While DCT's Type II error is initially larger than Fisher-z with large cardinalities in small sample size, it significantly decreases and matches the performance of others when the sample size reaches 500, showcasing DCT's effectiveness. This outcome aligns with our expectations, as the baselines do not adequately address the issues arising from discretization.
>
> >Q5: Why random boundaries?
>
> A5: Thanks for your question. We randomly set the boundaries to ensure that the experiment is sufficiently general. In real-world scenarios, we **do not have control** over where the discretization boundaries are located. To alleviate your concern, we conduct experiments testing the Type I and Type II error on data are equiprobable discretized.  Please kindly refer to A4 for a detailed experiment setting and analysis.

---

> > ### Comment · Reviewer_dVXS · 2024-08-12
> >
> > I thank the author for their rebuttal.
> > However, I will not change my assessment of the paper.
> > I am a bit confused by the test with calibration introduced in the rebuttal: in the original paper the test looked already correctly calibrated. If the new results with the calibrated tests have to be introduced in the paper, this is an important change, which in my view requires a resubmission of the paper.
> >
> > For future works, In the figures I suggest to make it easier to understand which method correspond to which line.

---

> ### Author Response · Authors · 2024-08-07
> **Some references**
>
> We hope that our response has answered your questions. If you need any more information, please feel free to contact us. We look forward to further discussions with you.
>
>
>
>
> [1] Tsamardinos, I., Brown, L. E., & Aliferis, C. F. (2006). The max-min hill-climbing Bayesian network structure learning algorithm. Machine learning, 65(1), 31-78.
>
> [2] Yishi Zhang, Zigang Zhang, Kaijun Liu, and Gangyi Qian. An improved iamb algorithm for markov blanket discovery. J. Comput., 5(11):1755–1761, 2010

---

> ### Author Response · Authors · 2024-08-12
> **Clarification about the Calibration**
>
> We are sorry for the confusion. We believe this is a misunderstanding that has already been addressed. We would sincerely appreciate it if the reviewers could reevaluate our paper given the following information:
>
> 1. **The figures are not contradictory; they have different meanings.**
>
>  -  In Fig.2 of the original paper, we evaluate the Type II error without calibrating (control) the Type I error (given the **nominated Type I error**). This approach is meaningful in practical scenarios, such as in causal discovery on real-world datasets, where tests are conducted directly on the variables of interest without access to synthetic datasets for calibration (need to use synthetic datasets to empirically determine the region corresponding to the desired significance level).
>
> - Figures included in the rebuttal compare Type II error of different methods under the same **empirical Type I error**. As illustrated in Fig.1 in the main text, the tested pairs are falsely concluded as d-connected given the discretized variables by baseline methods, leading to the issue that the calculated p-value is always close to zero. Thus, for a fair comparison, we evaluate the Type II error under calibration.
>
> - We would like to emphasize even in the evaluation of tests without calibration, Fig. 1 in the main text is already sufficient for demonstrating that DCT outperforms existing methods in the presence of discretization. Specifically, when the sample size is 2000, the Type I error of DCT is significantly smaller than other baseline methods and aligned with $\alpha$, and the Type II error becomes nearly identical. For all other baseline methods, the Type I error can not be controlled regardless of the sample size.
>
> 2. **The issue is minor and has been easily addressed.**
>
> - We **do not need to modify any figures or change any testing procedures; the only difference is the way of evaluation**. By simply including Figures in the rebuttal in the revised paper, this issue is addressed.

---

> ### Author Response · Authors · 2024-08-14
>
> Dear Reviewer dVXS,
>
> Thank you for your time and assistance in reviewing our submission. This is a kind reminder to inquire if your concerns about calibration have been addressed. Are there any confusions?
>
> Best regards,
>
> Authors of Submission3449

---

### Official Review · Reviewer_Q2XK · 2024-07-05

**Soundness:** 4
**Presentation:** 3
**Contribution:** 4
**Rating:** 9
**Confidence:** 5

**Summary:**

Authors propose a test for conditional independence in case of discretized variables, i.e. variables that originally were defined over a continuous domain and are then mapped to a discrete domain. In this case, a binary domain. Authors propose to bridge the unobserved continuous variables with the observed discretized variables with equations modeling the original covariance/precision matrix coefficients. Both theoretical and experimental evidence support the proposed testing  methods.

Typo at page 13, line 491, equation 17, missing closed bracket.

Citation 3, 4 are the same reference.

**Strengths:**

The paper contributes in a significant way on a topic which is crucial in the context of structure learning/causal discovery. Specifically, the proposed theoretical framework is solid and sound, explaining the logical steps that lead to the conditional independence test. The major strength points of this contribution are:

- The self-contained graphical representation of the discretized variables and their original continuous ones,
- The flexibility of the bridge equations, that can be adapted to specific cases without compromising the theoretical soundness.
- The performance of both the unconditional and the conditional independence test.

**Weaknesses:**

The only weakness is that I would do more experiments.

**Questions:**

- Where is the "vec" operator defined?

**Limitations:**

The only limitation, that is also discussed by authors, is that the "discretized" variables are in fact "binary" variables, which limits the applicability of the proposed test.

---

> ### Author Rebuttal · Authors · 2024-08-07
>
> >Q1: I would do more experiments
>
> A1: Thanks for your positive feedback. Based on the suggestions from reviewer otB5 and reviewer dvXS, we have supplemented additional experiments to more comprehensively evaluate the power of the proposed test in small sample sizes and included some additional baselines in comparison of Type I error and Type II error. They are shown in Figure1 and Figure2 in the attached pdf correspondingly.
>
> >Q2: Where is the "vec" operator defined.
>
> A2: Sorry for the confusion. Kindly refer to line587 in Appendix A 7.2. We will include its definition in the main text in the revised version.
>
> >Q3: Typos and same reference:
>
> A3: Thank you for your thorough review. We will correct this in the revised version.
>
> We are greatly encouraged by your response. If you have any questions, please do not hesitate to contact us. We look forward to further discussions with you.

---

> > ### Comment · Reviewer_Q2XK · 2024-08-11
> >
> > Thank you for your reply, I'm satisfied with the rebuttal, I'll keep my scores as they are.

---

> > > ### Author Response · Authors · 2024-08-11
> > > **Thank You for Your Acknowledgment**
> > >
> > > Dear Reviewer Q2XK,
> > >
> > > Thank you for your positive support and comments. We will carefully incorporate additional experiments in our revised version as suggested.
> > >
> > > Best regards,
> > >
> > > Submission3449 Authors

---

### Official Review · Reviewer_szaU · 2024-07-10

**Soundness:** 4
**Presentation:** 3
**Contribution:** 3
**Rating:** 7
**Confidence:** 4

**Summary:**

This paper presents a novel statistical method for testing conditional independence (CI) when some of the data is discretized. Initially, the authors introduce bridge equations to estimate covariance and establish asymptotic normality, facilitating an unconditional independence test. For the conditional independence test, they employ nodewise regression to recover precision coefficients. Theoretical analysis and empirical validation are provided to showcase the method’s effectiveness.

**Strengths:**

This paper introduces a conditional independence test tailored for scenarios with discretized data, which often encountered in financial analysis and healthcare due to data collection or measurement constraints. The CI test is highly adaptable, capable of handling situations where both variables are discretized, both are continuous, or one is discretized. Numerical experiments on both synthetic and real-world datasets demonstrate superior performance in various scenarios.

**Weaknesses:**

The development relies on the assumption of a multivariate Gaussian distribution, which is rather stringent.

**Questions:**

1. Are there any references that address the issue of discretization in conditional independence testing?

2. In line 157, the definition of the boundary $h_j$ essentially discretizes $X_j$ into two parts. I have two questions regarding this issue:
(a). If the original $X_j$ consists of more than two parts, will discretizing $X_j$ into only two parts cause some efficiency loss?
(b). There may be multiple ways to define $h_j$, such as replacing $E\tilde{X_j}$ with other quantities. Is there an optimal choice for $h_j$, given that it impacts the asymptotic variance of the estimated covariance according to Theorem 2.5?

**Limitations:**

The development relies on the assumption of a multivariate Gaussian distribution, which is rather stringent.

---

> ### Author Rebuttal · Authors · 2024-08-07
>
> >Q1: Assumption of a multivariate Gaussian distribution
>
> A1: Thank you for your valuable question. We acknowledge that the assumption of multivariate Gaussian distributions can limit the generality of the proposed test. However, we would like to share a few points regarding its reasonability:
> 1. **Challenges in Conditional Independence**: Inferring the conditional independence of latent variables based on their discretized values is indeed a complex problem. The discretization drastically reduces available information. Without a parametric assumption, establishing the statistics which reflects the real conditional independence and inferring its null distribution by only using those discretized values is particularly challenging and could even be overly ambitious. On the other hand, we hope this paper can inspire the community to propose more general and powerful solution to handle this obvious but overlooked spurious conditional dependence caused by discretization.
> 2. **Empirical  Performance**: Although the theory requires that the latent continuous variables follow a multivariate Gaussian distribution, we empirically validate the effectiveness of DCT even when the assumptions are violated. As demonstrated in **Appendix C.1** and **Figures 5 and 6** where we present causal discovery results for various distributions including linear uniform, linear student distribution, linear exponential, and nonlinear Gaussian distributions. From the experiment, DCT still shows superior performance than other baselines.
> 3. **Popularity of Copula Model**: The assumption of multivariate Gaussian, also called copula model, is well-studied and widely accepted in the community. There is a substantial body of work demonstrating the effectiveness of the copula model in various scenarios [1] [2] [ 3]. Technically, our model can be referred as a semiparametric Gaussian copula model, which can extend to elliptical distribution [4].
>
> >Q2: Are there any references that address the issue of discretization in conditional independence testing?
>
> A2: Thanks for your great questions. As far as we know, the most related work towards handling discretization is proposed by Fan [1], which also adopts the copula model. Compared with DCT, they use Kendall’s tau to reflect the independence relationship. However, their work still shorts by providing the statistical inference of interested parameters, i.e., a valid conditional independence test.
>
>
> >Q3: If the original $𝑋_𝑗$ consists of more than two parts, will discretizing $𝑋_𝑗$ into only two parts cause some efficiency loss?
>
> A3: You are totally right. In the current framework, the estimation of hidden covariance $\sigma_{j_1,j_2}$ is accomplished by solving a single bridge equation. This equation can be interpreted as looking for the suitable $\sigma_{j_1,j_2}$ to match the "region" in continuous domains, linking the covariance with discretized observations. However, there might be multiple bridge equations available while the current framework only allows the usage of one of them. This is, as discussed in the paper, by far the largest limitation. In the future, we will try to make efforts towards this problem to solve more bridge equations thus improving the sample efficiency.
>
> >Q4: Is there an optimal choice for $h_j$?
>
> A4: Thanks for your insightful question.  You are totally right that there are multiple ways to define $h_j$. We choose the mean of $E \tilde{X}_j$ as the quantity just for its simplicity. Currently, there is no theoretical progress to systematically determine the optimal $h_j$. However, we believe it is doable empirically. One naive solution would be to test $h_j$ with different quantiles ($E \tilde{X}_j$,  $2E \tilde{X}_j$, etc.), calculate the corresponding variance using Theorem~2.5 and choose the one with the smallest variance.
>
>
> -------------------------
> We believe our response has resolved your queries. If there are any more questions, please contact us at your convenience. We anticipate further conversations with you.
>
>
> [1] Fan, J., Liu, H., Ning, Y., and Zou, H. High dimensional semiparametric latent graphical model for mixed data.   Journal of the Royal Statistical Society Series B: Statistical Methodology, 79(2):405–421, 2017.
>
> [2] Zhang A, Fang J, Hu W, et al. A latent Gaussian copula model for mixed data analysis in brain imaging genetics[J]. IEEE/ACM transactions on computational biology and bioinformatics, 2019, 18(4): 1350-1360.
>
> [3] Liu H, Lafferty J, Wasserman L. The nonparanormal: semiparametric estimation of high dimensional undirected graphs[J]. Journal of Machine Learning Research, 2009, 10(10).
>
> [4] Barber, Rina Foygel and Mladen Kolar. “ROCKET: Robust Confidence Intervals via Kendall's Tau for Transelliptical Graphical Models.” _ArXiv_ abs/1502.07641 (2015): n. pag.

---

> > ### Comment · Reviewer_szaU · 2024-08-09
> >
> > Thank you for the rebuttal. I am satisfied with the response and will raise my score to 7.

---

> > > ### Author Response · Authors · 2024-08-11
> > > **Thank You for Your Acknowledgment**
> > >
> > > Dear Reviewer szaU,
> > >
> > > Thank you for your acknowledgment and efforts in reviewing our submission.
> > >
> > > Best regards,
> > >
> > > Submission3449 Authors

---

### Official Review · Reviewer_otB5 · 2024-07-14

**Soundness:** 3
**Presentation:** 3
**Contribution:** 2
**Rating:** 5
**Confidence:** 3

**Summary:**

The paper addresses a critical issue in Conditional Independence (CI) testing methods, specifically when the available data is a discretized version of the original continuous data. Traditional CI testing methods often assume that discretized observations can directly substitute for continuous variables, leading to erroneous conclusions. To overcome this limitation, the authors introduce a novel CI test tailored for discretized data. The key innovation lies in using a bridge equation and nodewise regression to estimate the precision coefficients that reflect CI relationships among latent continuous variables.

**Strengths:**

The paper tackles a highly relevant and important problem within the realm of statistical analysis and CI testing.
The paper is well-written and presents the concepts clearly.
The proposed method is novel, and the theoretical contributions are solid, providing a robust foundation for CI testing in discretized data settings.

**Weaknesses:**

Assumption of Multivariate Normality: A primary limitation is the assumption that the data follow a multivariate normal distribution. This assumption simplifies the derivation of bridge equations for unconditional independence testing and the use of nodewise regression for the CI test. However, it restricts the applicability of the method to this specific class of variables. It is unclear how the method would perform with unknown or non-normal variables.

Discretization Modeling: The paper models discretization as a binarization operation applied to observed variables. This assumption may not hold in all practical scenarios. The performance of the proposed method on datasets with different types of discretization (beyond binarization) remains unexamined and is an important consideration for real-world applications.

Empirical Results: According to the empirical results, the proposed Discretized CI Test (DCT) shows smaller power compared to baseline methods. This indicates that while the method is innovative, its practical effectiveness in terms of power may be limited in some scenarios.

**Questions:**

See above comments.

**Limitations:**

See above comments.

---

> ### Author Rebuttal · Authors · 2024-08-07
>
> >Q1 : Assumption of Multivariate Normality, How would the method perform with unkown or non-normal variables
>
> A1: Thank you for your question. We appreciate your insightful feedback. We acknowledge that the assumption of multivariate Gaussian distributions can limit the generality of the proposed test. However, we would like to share a few points regarding its reasonability:
> 1. **Challenges in Conditional Independence**: Inferring the conditional independence of latent variables based on their discretized values is indeed a complex problem. The discretization drastically reduces available information. Without mild assumptions, establishing the statistics which reflects the real conditional independence and inferring its null distribution by only using those discretized values is particularly challenging and could even be overly ambitious. On the other hand, we hope this paper can inspire the community to propose more general and powerful solution to handle this obvious but overlooked spurious conditional dependence caused by discretization.
> 2. **Empirical  Performance**: Although the theory requires that the latent continuous variables follow a multivariate Gaussian distribution, we empirically validate the effectiveness of DCT even when the assumptions are violated. As demonstrated in **Appendix C.1** and **Figures 5 and 6** where we present causal discovery results for various distributions including linear uniform, linear student distribution, linear exponential, and nonlinear Gaussian distributions. From the experiment, DCT still shows superior performance than other baselines.
> 3. **Popularity of Copula Model**:  The assumption of multivariate Gaussian, also called Gaussian copula model, is well-studied and widely accepted in the community. There is a substantial body of work demonstrating the effectiveness of the copula model in various scenarios [1] [2] [ 3]. Technically, our model can be referred as a semiparametric Gaussian copula model, which can extend to elliptical distribution [4].
>
> >Q2: Discretization Modelling
>
> A2: Thanks for your question and we would like to appreciate the opportunity to clarify our approach. Our method indeed treats the discretized observations as a binary variable to provide a unified framework for all types of data. However, binarization is technically used as data processing rather than the nature of data. That is, binarization is an **operation** rather than **assumption**. Theoretically, any discrete variable can be further discretized into a binary variable. Specifically, even the observed variable $\tilde{X}_j$ can have multiple values, we binarize it with its sample mean as the boundary. We then use this produced binary variable to conduct the statistical inference of the precision matrix of original continuous variables.
>
> The binarization operation, of course, wastes the available information and causes some efficiency loss. However, it embarks us to propose methods of handling data more effectively. In the future, we will try to use as much information, i.e., more equations to determine the interested parameters.
>
> Despite treating discrete variables as binary, we empirically demonstrate the superiority of DCT compared to baselines (Fisher-z, Chi-square) that handle discrete variables directly. Please refer to **Section 3.1** and **Figure 2**, where data discretized into multiple values (K=2, 4, 8, 12) are evaluated.
>
> >Q3: The performance of the proposed method on datasets with different types of discretization (beyond binarization) remains unexamined.
>
> A3: Please kindly refer to **Section 3.1** and **Figure 2**, where we empirically demonstrate the superiority of DCT compared to baselines (Fisher-z, Chi-square) that handle discrete variables directly(The cardinality of discretization K=2, 4, 8, 12).
>
> >Q4: Small power
>
> A4: We have also observed that the power of the DCT in experiments can be relatively low compared with a part of baselines when the **sample size is small**. However, we must point out that this is the result **without calibration**.
>
> Just as the cases illustrated in Figure1, the tested pairs are conditional dependent without given the discretized variables in principle. This leads to the p-value obtained from conventional tests always being close to 0. In such cases, directly comparing the obtained p-value with the preset significance level ($\alpha=0.05$) is unfair for DCT. Therefore, we provide the results of the test with **calibration** applied to cases with relatively small data samples ($n=100,500$)  in Figure1 in the attached pdf. Specifically, we empirically determine the value corresponding to the five percent quantile under the $H_0$. We then use this value as the threshold to determine the rejection of $H_0$ in the evaluation of Type II error for a fair comparison.
>
> From the experiment result, DCT exhibits superior performance compared with Chi-square tests applied to every kinds of discrete data and Fisher-z test applied to binary data.  As the cardinality of discretization increases, the performance of DCT does not match that of Fisher-z test. This result is not surprising as the discretization drastically reduces the available information. Additionally, we need to note that all other baselines maintain significantly higher Type I error rates as shown in Figure2 of the main text. At the same time, when the sample size increases,  Type II error dramatically decreases while maintaining ideal Type I error, demonstrating the efficacy of the proposed approach. However, for the baselines, although the Type II error is further reduced, the Type I error increases significantly. This highlights the necessity of developing tests that correctly infer conditional independence, as achieved in this paper.
>
> ------
> We hope our response has addressed your questions. Please feel free to contact us if you have any further inquiries. We look forward to further discussions with you.

---

> ### Author Response · Authors · 2024-08-07
> **Some references**
>
> [1] Fan, J., Liu, H., Ning, Y., and Zou, H. High dimensional semiparametric latent graphical model for mixed data.   Journal of the Royal Statistical Society Series B: Statistical Methodology, 79(2):405–421, 2017.
>
> [2] Zhang A, Fang J, Hu W, et al. A latent Gaussian copula model for mixed data analysis in brain imaging genetics[J]. IEEE/ACM transactions on computational biology and bioinformatics, 2019, 18(4): 1350-1360.
>
> [3] Liu H, Lafferty J, Wasserman L. The nonparanormal: semiparametric estimation of high dimensional undirected graphs[J]. Journal of Machine Learning Research, 2009, 10(10).
>
> [4] Barber, Rina Foygel and Mladen Kolar. “ROCKET: Robust Confidence Intervals via Kendall's Tau for Transelliptical Graphical Models.” _ArXiv_ abs/1502.07641 (2015): n. pag.

---

### Author Rebuttal · Authors · 2024-08-07

Dear **Reviewer otB5, szaU, Q2XK, dVXS**, We deeply appreciate the time and effort you have invested in evaluating our work. Your insightful feedback has significantly contributed to improving the quality of our paper.  It's encouraging that **Reviewer otB5, Q2XK** acknowledge we are targeting a **very important problem**, **Reviewer otB5, Q2XK** acknowledge our **solid theoretical results**, **Reviewer Q2XK, szaU** think that our method is highly **flexible**, and **Reviewer otB5, dVXS** acknowledge the **originality** of our method.

Here we provide a general response to summarize our rebuttal.


$\circ$ To Reviewer otB5 and Reviewer szaU, we acknowledge that the assumption may limit the generality of applications while we argue its reasonability based on its feasibility and practical performance.

$\circ$ To Reviewer otB5 and Reviewer dVXS, we clarify the presented power is a result without calibration, which is an **unfair** comparison. We presented the result with calibration in the attached pdf.

$\circ$ To Reviewer otB5, we clarify the binarization operation which is independent with the nature of data.

$\circ$ To Reviewer szaU, we discuss the efficiency loss using binarization and the possible optimal choice of $h_j$.

$\circ$ To Reviewer Q2XK, we add more experiments to more comprehensively evaluate the power and more baselines for comparison.

$\circ$ To Reviewer dVXS, we emphasize the motivation and stress the fundamental difference between  DCT with traditional tests.

$\circ$ To Reviewer dVXS, we conduct the experiment where data are discretized equiprobably and compare DCT with old and new baselines.

Thanks again for your contribution. We hope our explanation provides the clarity you were seeking. Please feel free to reach out if further clarification is required. Your insights are greatly appreciated.

Best Regards, Authors of Submission3449.

---

### Decision · Program_Chairs · 2024-09-25

**Decision:**

Reject

**Comment:**

The paper introduces a novel method for testing conditional independence when some variables are discretized, a frequent challenge in real-world data analysis. The method builds on bridge equations and nodewise regression to infer relationships among latent continuous variables, aiming to mitigate potential misjudgments of conditional independence due to discretization.

While the approach addresses a relevant problem and is built on strong theoretical foundations, the reviewers had mixed opinions. Some raised significant concerns, including the assumption of an unrealistic multivariate Gaussian distribution, issues with test calibration, and inadequate comparisons with modern conditional independence tests.

Although the authors attempted to address these concerns, some issues remain unresolved. Overall, the paper presents a novel contribution to a relevant problem, but the authors need to make a more thorough effort to better address these concerns. Given the highly competitive nature of this conference, I cannot recommend accepting the paper in its current form. I strongly encourage the authors to revise the manuscript and consider submitting it to another venue.